# Loss of dihydroceramide desaturase drives neurodegeneration by disrupting endoplasmic reticulum and lipid droplet homeostasis in glial cells

Yuqing Zhu[1], Kevin Cho[2,3,4], Haluk Lacin[5], Yi Zhu[1], Jose T DiPaola[1], Beth A Wilson[1], Gary Patti[2,3,4], James B Skeath[1]*

[1]Department of Genetics, Washington University School of Medicine, St. Louis, United States; [2]Department of Chemistry, Washington University in St. Louis, St. Louis, United States; [3]Department of Medicine, Washington University School of Medicine, St. Louis, United States; [4]Center for Mass Spectrometry and Metabolic Tracing, Washington University in St. Louis, St. Louis, United States; [5]Division of Biological and Biomedical Systems, University of Missouri-Kansas City, Kansas City, United States

*For correspondence:
jskeath@wustl.edu

## eLife Assessment

This study on the loss of DEGS1 in the developing larval brain **convincingly** shows the accumulation of dihydroceramide in the CNS which induces severe alterations in the morphology of glial subtypes as well as a reduction in glial number. The localization of DEGS1/ifc primarily to the ER is also **compelling** and interesting, and the loss of DEGS1/ifc clearly drives ER expansion and reduces the levels of TGs. This is an **important** contribution to the role of lipid metabolism in neural development and disease.

**Abstract** Dihydroceramide desaturases convert dihydroceramides to ceramides, the precursors of all complex sphingolipids. Reduction of DEGS1 dihydroceramide desaturase function causes pediatric neurodegenerative disorder hypomyelinating leukodystrophy-18 (HLD-18). We discovered that *infertile crescent* (*ifc*), the *Drosophila DEGS1* homolog, is expressed primarily in glial cells to promote CNS development by guarding against neurodegeneration. Loss of *ifc* causes massive dihydroceramide accumulation and severe morphological defects in cortex glia, including endoplasmic reticulum (ER) expansion, failure of neuronal ensheathment, and lipid droplet depletion. RNAi knockdown of the upstream ceramide synthase *schlank* in glia of *ifc* mutants rescues ER expansion, suggesting dihydroceramide accumulation in the ER drives this phenotype. RNAi knockdown of *ifc* in glia but not neurons drives neuronal cell death, suggesting that *ifc* function in glia promotes neuronal survival. Our work identifies glia as the primary site of disease progression in HLD-18 and may inform on juvenile forms of ALS, which also feature elevated dihydroceramide levels.

## Introduction

Sphingolipids are key structural and functional components of the cell membrane in all eukaryotic cells and are enriched in glial cells, such as oligodendrocytes, where they comprise up to 30% of all membrane lipids (*Jackman et al., 2009*). All complex sphingolipids, like glycosylsphingolipids and

**eLife digest** Neurodegenerative diseases affect around 50 million people worldwide. They arise when neurons deteriorate and die. Neurodegeneration was thought to result from defects within neurons. But recent studies have shown that changes in brain cells known as glial cells – which surround, protect and nourish neurons – can also trigger this process.

The fat composition of the surrounding plasma membrane of glial cells differs from neurons and contains high levels of sphingolipids. These lipids regulate membrane fluidity – the movement of molecules within and through the membrane – and are also critical for cell signaling and the formation of nerve-insulating myelin sheaths.

All complex sphingolipids, such as sphingomyelins and gangliosides, are derived from ceramide. Enzymes called DEGS1 produce ceramide from dihydroceramide in the endoplasmic reticulum. Ceramides are then transported to the Golgi complex, where they are modified into complex sphingolipids.

In humans, mutations in the gene encoding DEGS1 cause a loss of the myelin sheath leading to a fatal neurodegenerative condition in children called hypomyelinating leukodystrophy-18. So far, it was unclear whether the accumulation of dihydroceramide or the depletion of ceramide might alter the function of neurons and glia enough to trigger neurodegeneration.

Zhu et al. addressed this question using genetically modified fruit fly larvae that lacked the DEGS1 gene. They discovered that in fruit flies, DEGS1 protects the nervous system from neurodegeneration by supporting the development and function of glial cells. In flies that lacked the gene, dihydroceramide accumulated in the central nervous system, which enlarged the endoplasmic reticulum in glial cells, causing them to swell. These morphological defects inhibited their ability to enwrap the cell bodies and axons of neurons with a supporting glial sheath. This suggests that a faulty DEGS1 gene may drive neurodegeneration as a secondary consequence of glial dysfunction.

By establishing a simple model system, Zhu et al. provide insight into how glial cells may contribute to neurodegeneration. Their results indicate that DEGS1 loss causes structural and functional defects in glial cells, preventing them from supporting neurons and ultimately leading to neurodegeneration. Because the ceramide synthesis pathway is conserved between fruit flies and humans, similar mechanisms likely contribute to neuronal degeneration in patients with DEGS1 mutations. A deeper understanding of these pathways could help identify strategies to slow the progression of hypomyelinating leukodystrophy-18 and open new avenues for therapy.

sphingomyelin, derive from ceramide (*Ghosh et al., 2013*; *Vacaru et al., 2013*; *Hannun and Obeid, 2018*). In the de novo ceramide biosynthesis pathway, dihydroceramide desaturases, such as DEGS1, produce ceramide from dihydroceramide by catalyzing the formation of a *trans* double bond between carbons 4 and 5 of the sphingoid backbone, which enhances conformational plasticity (*Li et al., 2002*; *Yasuda et al., 2016*). In humans, bi-allelic mutations in *DEGS1* cause hypomyelinating leukodystrophy-18 (HLD-18), a progressive, often fatal pediatric neurodegenerative disease marked by cerebral atrophy, white matter reduction, and hypomyelination (*Dolgin et al., 2019*; *Karsai et al., 2019*; *Pant et al., 2019*). The primary neural cell type impacted by loss of *DEGS1* function and the cell biology of how disruption of ceramide synthesis leads to neurodegeneration remain unknown.

The de novo ceramide biosynthetic pathway is well conserved among higher metazoans (*Hannun and Obeid, 2018*; *Fyrst et al., 2004*; *Acharya and Acharya, 2005*; *Dunn et al., 2019*; *Pan et al., 2023*). De novo ceramide synthesis occurs in the endoplasmic reticulum (ER) and starts with the rate-limiting activity of the serine palmitoyltransferase (SPT) complex, which condenses serine and palmitoyl-CoA (lauoryl-CoA in flies) to form 3-Ketosphinganine, which is converted to sphinganine by 3-Ketosphinganine reductase. Ceramide synthases condense sphinganine with acyl-CoA to generate dihydroceramide, which is converted to ceramide by dihydroceramide desaturases. Ceramide is then efficiently transported by the specific ceramide transporter CERT from the ER to the Golgi (*Kumagai et al., 2005*), where it undergoes headgroup modifications to produce complex sphingolipids that eventually translocate to the plasma membrane (*Kobayashi and Menon, 2018*). Mutations in most members of the SPT complex, ceramide synthases, and DEGS1 lead to neurodegeneration (*Hannun and Obeid, 2018*; *Dunn et al., 2019*; *Pan et al., 2023*), identifying the de novo ceramide biosynthesis pathway as a hotspot for neurodegenerative disease mutations.

Consistent with its central role in ceramide biogenesis, reduction or loss of *DEGS1* function in human patients or cell lines, mice, zebrafish, and flies drives dihydroceramide accumulation and ceramide depletion (*Karsai et al., 2019*; *Pant et al., 2019*; *Holland et al., 2007*; *Jung et al., 2017*). HLD-18 patients display reduced myelin sheath thickness in peripheral nerves, and knockdown of *DEGS1* function in zebrafish reduces the number of myelin basic protein-positive oligodendrocytes (*Karsai et al., 2019*; *Pant et al., 2019*), suggesting *DEGS1* regulates Schwann cell and oligodendrocyte development. In *Drosophila*, genetic ablation of *infertile crescent* (*ifc*), the fly *DEGS1* ortholog, drives activity-dependent photoreceptor degeneration (*Jung et al., 2017*), suggesting that *ifc* function is crucial for neuronal homeostasis. In addition, forced expression of a wild-type *ifc* transgene in neurons, glia, or muscles was shown to rescue the *ifc* mutant phenotype (*Jung et al., 2017*). Whether *ifc/DEGS1* acts primarily in glia, neurons, or other cells to regulate nervous system development then remains to be determined.

Research on *DEGS1* points to dihydroceramide accumulation as the driver of nervous system defects caused by *DEGS1* deficiency. Pharmacological or genetic inhibition of ceramide synthase function, which should reduce dihydroceramide levels, suppresses the observed reduction of MBP-positive oligodendrocytes in zebrafish and the activity-dependent photoreceptor degeneration in flies triggered by reduction of *DEGS1/ifc* function (*Pant et al., 2019*; *Jung et al., 2017*). Dihydroceramide accumulation may also contribute to other neurodegenerative diseases, as gain of function mutations in components of the SPT complex cause juvenile amyotrophic lateral sclerosis due to elevated sphingolipid biosynthesis, with dihydroceramides showing the greatest relative increase of all sphingolipids (*Lone et al., 2022*; *Lone et al., 2023*; *Syeda et al., 2024*; *Dohrn et al., 2024*; *Srivastava et al., 2023*). How dihydroceramide accumulation alters the cell biology of neurons, glia, or both to trigger myelination defects and neurodegeneration is unclear.

With its vast genetic toolkit, *Drosophila* is a powerful system in which to dissect the genes and pathways that regulate glial development and function (*Freeman and Doherty, 2006*; *Coutinho-Budd and Freeman, 2013*). In flies, six morphologically and functionally distinct glial subtypes regulate nervous system development and homeostasis (*Yildirim et al., 2019*; *Corty and Coutinho-Budd, 2023*). The surface perineurial and subperineurial glia act as physiochemical barriers to protect the nervous system and control metabolite exchange with the hemolymph, carrying out a similar function as the human blood–brain barrier (*Stork et al., 2008*; *Volkenhoff et al., 2015*). Residing between the surface glia and the neuropil, cortex glia ensheathe the cell bodies of neuroblasts and neurons in the CNS in protective, nutritive, honeycomb-like membrane sheaths. Ensheathing glia define the boundary of the neuropil and insulate axons, dendrites, and synapses from neuronal cell bodies in the CNS (*Freeman, 2015*). Astrocyte-like glia extend fine membrane protrusions that infiltrate the neuropil and form a meshwork of cellular processes that ensheathe synapses and regulate synaptic homeostasis in the CNS (*Freeman, 2015*). In the PNS, wrapping glia reside internally to the surface glia and insulate axons in the PNS to enhance neuronal signaling and provide energy support.

Using the *Drosophila* model, we found that *ifc* acts primarily in glia to regulate CNS development, with its loss disrupting glial morphology. Our work supports a model in which inappropriate accumulation and retention of dihydroceramide in the ER drives ER expansion, glial swelling, and the failure of glia to enwrap neurons, ultimately leading to neuronal degeneration as a secondary consequence of glial dysfunction. Given the conserved nature of de novo ceramide biosynthesis, our findings likely illuminate the exact mechanism through which elevated dihydroceramide levels drive neuronal degeneration and cell death in flies and humans.

## Results

### *ifc* contributes to the regulation of developmental timing and CNS structure

In an EMS-based genetic screen, we uncovered three non-complementing mutations that, when homozygous or trans-heterozygous to each other, resulted in identical phenotypes, including a 3-day or greater delay in reaching the late-third larval instar stage, reduced brain size, progressive ventral nerve cord elongation, axonal swelling, and lethality at the late larval or early pupal stage (*Figure 1A*; data not shown). Whole-genome sequencing revealed that *ifc* was independently mutated in each line: *ifc^{is1}* and *ifc^{is2}* encode V276D and G257S missense mutations, respectively, and *ifc^{is3}* encodes a

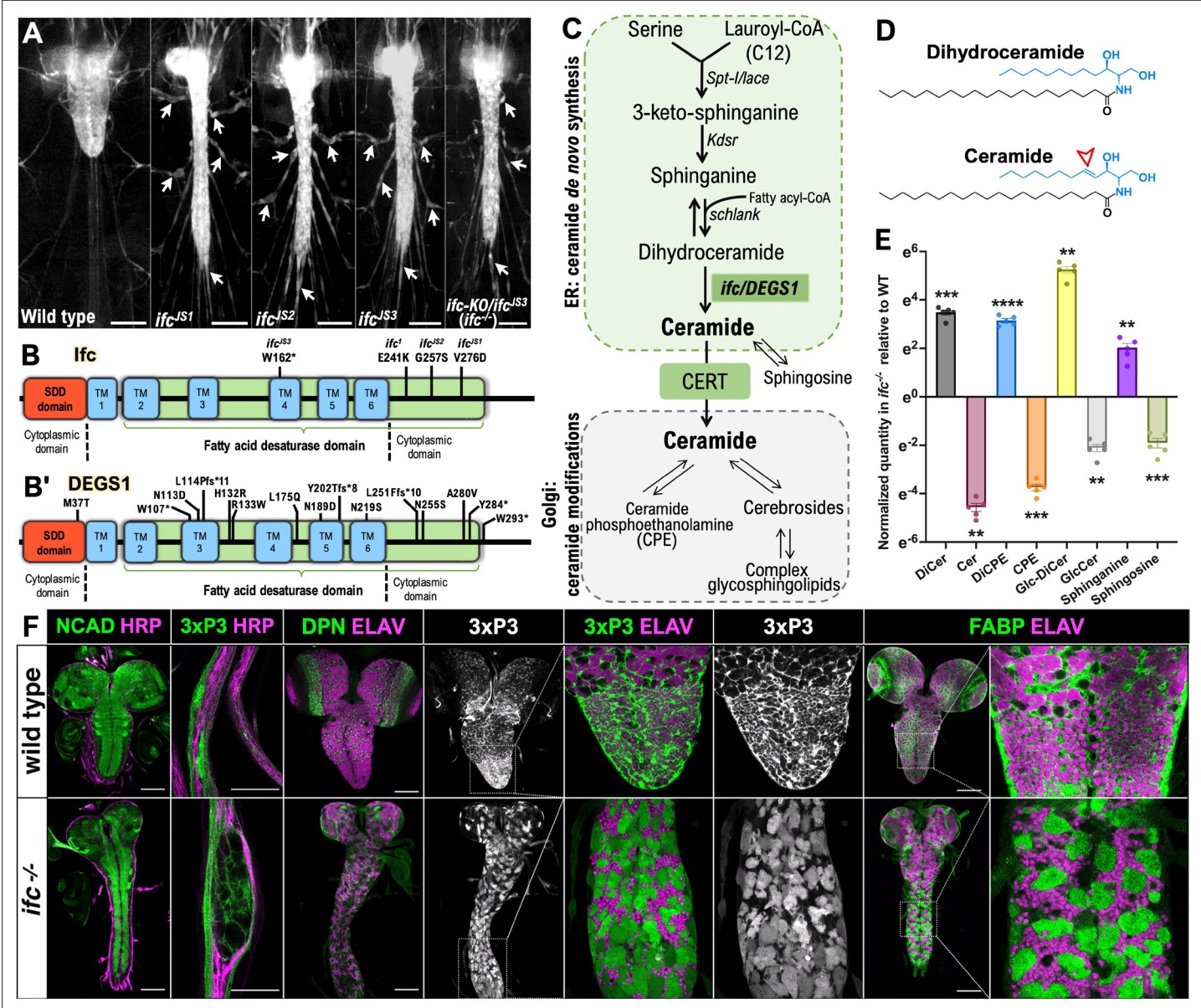

**Figure 1.** *ifc* regulates CNS and glial morphology. (**A**) Ventral views of late-third instar larvae of indicated genotype showing 3xP3 RFP labeling of CNS and nerves. Arrowheads indicate nerve bulges; scale bar is 200 µm. Schematic of Ifc (**B**) and human DEGS1 (**B′**) proteins indicating location and nature of *ifc* mutations and 15 HLD-18-causing *DEGS1* mutations (***Dolgin et al., 2019***; ***Karsai et al., 2019***; ***Pant et al., 2019***). (**C**) Schematic of de novo ceramide biosynthesis pathway indicating the subcellular location of ceramide synthesis and ceramide modifications. (**D**) Chemical structure of dihydroceramide and ceramide; arrow indicates *trans* carbon–carbon double bond between C4 and C5 in the sphingoid backbone created by the enzymatic action of Ifc/DEGS1. (**E**) Normalized quantification of the relative levels of dihydroceramide, ceramide, and six related sphingolipid species in the dissected CNS of wild-type and *ifc⁻/⁻* late-third instar larvae. (**F**) Ventral views of *Drosophila* CNS and peripheral nerves in wild-type and *ifc⁻/⁻* mutant late-third instar larvae labeled for NCAD to mark the neuropil, HRP to label axons, RFP to label glia, Dpn to label neuroblasts, ELAV to label neurons, and fatty acid binding protein (FABP) to label cortex glia. Anterior is up; scale bar is 100 µm for whole CNS images and 20 µm for peripheral nerve image. Statistics: **p < 0.01, ***p < 0.001, ****p < 0.0001.

The online version of this article includes the following source data and figure supplement(s) for figure 1:

**Figure supplement 1.** Ceramide metabolic changes in whole larvae.

**Figure supplement 2.** The *M{3xP3-RFP.attP}ZH-51D* transgene insert labels most glia.

**Figure supplement 3.** Most 3xP3-GFP or 3xP3-RFP transgenes label glia.

**Figure supplement 3—source data 1.** All lines tested.

**Figure supplement 4.** Newly generated fatty acid binding protein (FABP) antibody labels cortex glia.

*Figure 1 continued on next page*

*Figure 1 continued*

**Figure supplement 5.** Loss of *ifc* function reduces optic lobe size and optic lobe neuroblast number.

**Figure supplement 6.** Loss of *ifc* increases RFP fluorescence and induces the formation of bright RFP-positive puncta or aggregates.

**Figure supplement 7.** *ifc^{is1}* and *ifc^{is2}* alleles drive cortex glia swelling, ER expansion in cortex glia, and neuronal cell death.

W162* nonsense mutation (*Figure 1A, B*). Sanger sequencing also uncovered a missense mutation, E241K, in the molecularly uncharacterized *ifc* (*Jackman et al., 2009*) allele (*Figure 1B*). All four mutations reside in the fatty acid desaturase domain, the hotspot for mutations in human *DEGS1* that cause HLD-18 (*Figure 1B, B'*).

As a prior study reported that a CRISPR-generated, gene-specific deletion of *ifc*, *ifc-KO*, resulted in early larval lethality (*Jung et al., 2017*), we first confirmed the genetic nature of our *ifc* alleles because they caused late larval-early pupal lethality. Complementation crosses of each *ifc* allele against a deficiency of the region (Df(2L)BSC184) and *ifc-KO* revealed that all combinations, including larvae trans-heterozygous for *ifc-KO* over Df(2L)BSC184, survived to the late larval-early pupal stage and yielded phenotypes identical to those detailed above for the newly uncovered *ifc* alleles. Flies homozygous for *ifc-KO*, however, died as early larvae. Further analysis uncovered second site mutation(s) in the 21E2 chromosomal region responsible for the early lethal phenotype of the *ifc-KO* chromosome (see Methods). When uncoupled from these mutation(s), larvae homozygous for the 'clean' *ifc-KO* chromosome developed to the late larval-early pupal stage and manifested phenotypes identical to the other *ifc* alleles. This analysis defined the correct lethal phase for *ifc* and identified our *ifc* alleles as strong loss of function mutations.

## Loss of *ifc* function drives ceramide depletion and dihydroceramide accumulation

As *ifc/DEGS1* converts dihydroceramide to ceramide, we used untargeted lipidomics on whole larvae and the isolated CNS from wild-type and *ifc-KO/ifc^{JS3}* larvae (hereafter termed *ifc^{−/−}* larvae) to assess the effect of loss of *ifc* function on metabolites in the ceramide pathway (*Figure 1C*). Loss of *ifc* function resulted in a near complete loss of ceramides and a commensurate increase in dihydroceramides in the CNS and whole larvae (*Figure 1E*, *Figure 1—figure supplement 1*). Sphinganine, the metabolite directly upstream of dihydroceramide, also exhibited a significant increase in its levels in the absence of *ifc* function, while metabolites further upstream were unchanged in abundance or undetectable (*Figure 1E*, *Figure 1—figure supplement 1*). Ceramide derivatives like sphingosine, CPE, and glucosyl-ceramide (Gl-Cer) were reduced in levels and replaced by their cognate dihydroceramide forms (e.g., Glc-DiCer) (*Figure 1E*, *Figure 1—figure supplement 1*). Loss of the enzymatic function of Ifc then drives dihydroceramide accumulation and ceramide loss.

## *ifc* governs glial morphology and survival

To connect this metabolic profile to a cellular phenotype, we assayed *ifc* function in the CNS. We leveraged the expression of fatty acid binding protein (FABP) as a marker of cortex glia (*Kis et al., 2015*) and that of the *M{3xP3-RFP.attP}* phi-C31 'landing pad' transgene, which resided in the isogenic target chromosome of our screen, as a marker of most glia (*Figure 1—figure supplements 2–3*). In addition, we labeled neuroblasts with Deadpan (Dpn), neurons with ELAV, and axons with N-Cadherin (NCad). In *ifc^{−/−}* larvae, we observed a clear reduction in Dpn-positive neuroblasts in the optic lobe, swelling of glia in peripheral nerves, enhanced RFP expression in the CNS, and the presence of large swollen, cortex glia identified by RFP labeling and FABP expression (*Figure 1F*, *Figure 1—figure supplements 4 and 5*). In wild-type larvae, cortex glia display compact cell bodies and fully enwrap individual neuronal cell bodies with their membrane sheaths (*Figure 1F*, *Figure 1—figure supplement 4*; *Kis et al., 2015*). In *ifc^{−/−}* larvae, cortex glia display swollen cell bodies, fail to fully enwrap neuronal cell bodies, displace neurons from their regular arrangement, and appear to contain brightly fluorescent RFP-positive aggregates (*Figure 1F*, *Figure 1—figure supplement 6*). *ifc* is then necessary for glial development and function in the larval nervous system. We observed identical CNS phenotypes in larvae homozygous mutant for the *ifc^{is1}* and *ifc^{is2}* alleles (*Figure 1—figure supplement 7*).

To track the impact of *ifc* on glial morphology, we combined GAL4 lines specific for each glial subtype with a UAS-linked membrane-tagged GFP transgene (Myr-GFP) and the MultiColor FlpOut

system (*Kremer et al., 2017*; *Pfeiffer et al., 2010*; *Nern et al., 2015*). Using this approach, we determined that loss of *ifc* function affects all CNS glial subtypes except perineurial glia (*Figure 2E, E', J, J'*). Cortex glia appeared swollen, failed to enwrap neurons, and accumulated large amounts of Myr-GFP$^+$ internal membranes (*Figure 2A, A', F, F'*). Ensheathing glia (*Figure 2B, B', G, G'*), and subperineurial glia (*Figure 2D, D', I, I'*) also displayed swollen, disorganized cell bodies and accumulated Myr-GFP$^+$ internal membranes. Astrocyte-like glia displayed smaller cell bodies, reduced membrane extensions, and disrupted organization along the dorsal–ventral nerve cord (*Figure 2C, C'*). We conclude that *ifc* regulates the morphology of most glial subtypes in the larval CNS.

Next, we asked if loss of *ifc* function alters the number of each glial subtype. Using the same glial subtype-specific GAL4 lines to drive a nuclear-localized GFP transgene, we counted the total number of all CNS glial subtypes, except perineurial glia, in wild-type and *ifc*$^{-/-}$ larvae. To remove the small size of the brain in *ifc*$^{-/-}$ larvae as a confounding factor, we focused our analysis on the ventral nerve cord. The number of subperineurial glia was unchanged between the two genotypes, but we observed a 12%, 40%, and 72% reduction in the number of astrocyte-like, ensheathing, and cortex glia, respectively, in *ifc*$^{-/-}$ larvae relative to wild-type (*Figure 2K–N*). Our data reveal a broad role for *ifc* in regulating glial cell morphology and number in the *Drosophila* larval CNS. Subsequent experiments revealed that a reduction in cell proliferation and an increase in apoptosis both contribute to the observed reduction in the number of cortex glia (*Figure 2—figure supplement 1*).

## *ifc* acts in glia to regulate glial and CNS development

Prior work in zebrafish showed that *DEGS1* knockdown reduced the number of myelin basic protein-positive oligodendrocytes (*Pant et al., 2019*); in flies, loss of *ifc* function in the eye drove photoreceptor degeneration (*Jung et al., 2017*). Neither study uncovered the cell type in which *ifc/DEGS1* acts to regulate neural development. To address this question, we used the GAL4/UAS system, RNAi-mediated gene depletion, and gene rescue approaches to see if *ifc* acts in neurons or glia to control glial development and CNS morphology. First, we used a UAS-linked *ifc*-RNAi transgene to deplete *ifc* function in all neurons (*elav-GAL4; repo-GAL80*) or all glia (*repo-GAL4*). Focusing on FABP-positive cortex glia due to their easily scorable phenotype, we found that pan-glial, but not pan-neuronal, knockdown of *ifc* recapitulated the swollen cortex glia phenotype observed in *ifc* mutant larvae (*Figure 3C, D, J*).

To complement our RNAi approach, we asked if GAL4-driven expression of a wild-type *Drosophila ifc* or human *DEGS1* transgene rescued the *ifc*$^{-/-}$ CNS phenotype. In the absence of a GAL4 driver, the *ifc* transgene drove weak rescue of the cortex glia phenotype (*Figure 3E, K*), consistent with modest GAL4-independent transgene expression reported for UAS-linked transgenes (*Southall et al., 2013*). Pan-neuronal expression of *ifc* drove modest rescue of the *ifc* CNS phenotype beyond that observed for the *ifc* transgene alone (*Figure 3F, K*), but pan-glial expression of *ifc* fully rescued the *ifc* mutant cortex glia phenotype and other CNS phenotypes (*Figure 3H, K*, *Figure 3—figure supplement 1*). Identical experiments using the human *DEGS1* transgene revealed that only pan-glial *DEGS1* expression provided rescuing activity, albeit at much weaker levels than the *Drosophila ifc* transgene (*Figure 3G, I, K*). Pan-glial expression of the *Drosophila ifc* transgene was, in fact, ~15-fold more potent than the human *DEGS1* transgene at rescuing the *ifc* lethal phenotype to adulthood: When *ifc* was expressed in all glia, 57.9% of otherwise *ifc* mutant flies survived to adulthood (*n* = 2452), but when *ifc* was replaced by *DEGS1* only 3.9% of otherwise *ifc* mutant flies reached adulthood (*n* = 1303). No *ifc* mutant larvae reached adulthood in the absence of either transgene (*n* = 1030). We infer that *ifc* acts primarily in glia to govern CNS development and that human *DEGS1* can partially substitute for *ifc* function in flies despite a difference in the preferred length of the sphingoid backbone in flies versus mammals (*Fyrst et al., 2004*).

Results of our gene rescue experiments conflict with a prior study on *ifc* in which expression of *ifc* in neurons was found to rescue the *ifc* phenotype (*Jung et al., 2017*). In this context, we note that *elav-GAL4* drives UAS-linked transgene expression not just in neurons, but also in glia at appreciable levels (*Berger et al., 2007*; *Lacin et al., 2024*) and thus needs to be paired with *repo-GAL80* to restrict GAL4-mediated gene expression to neurons. Thus, 'off-target' expression in glial cells may account for the discrepant results. It is, however, more difficult to reconcile how neuronal or glial expression of *ifc* would rescue the observed lethality of the *ifc-KO* chromosome given the presence of additional lethal mutations in the 21E2 region of the second chromosome.

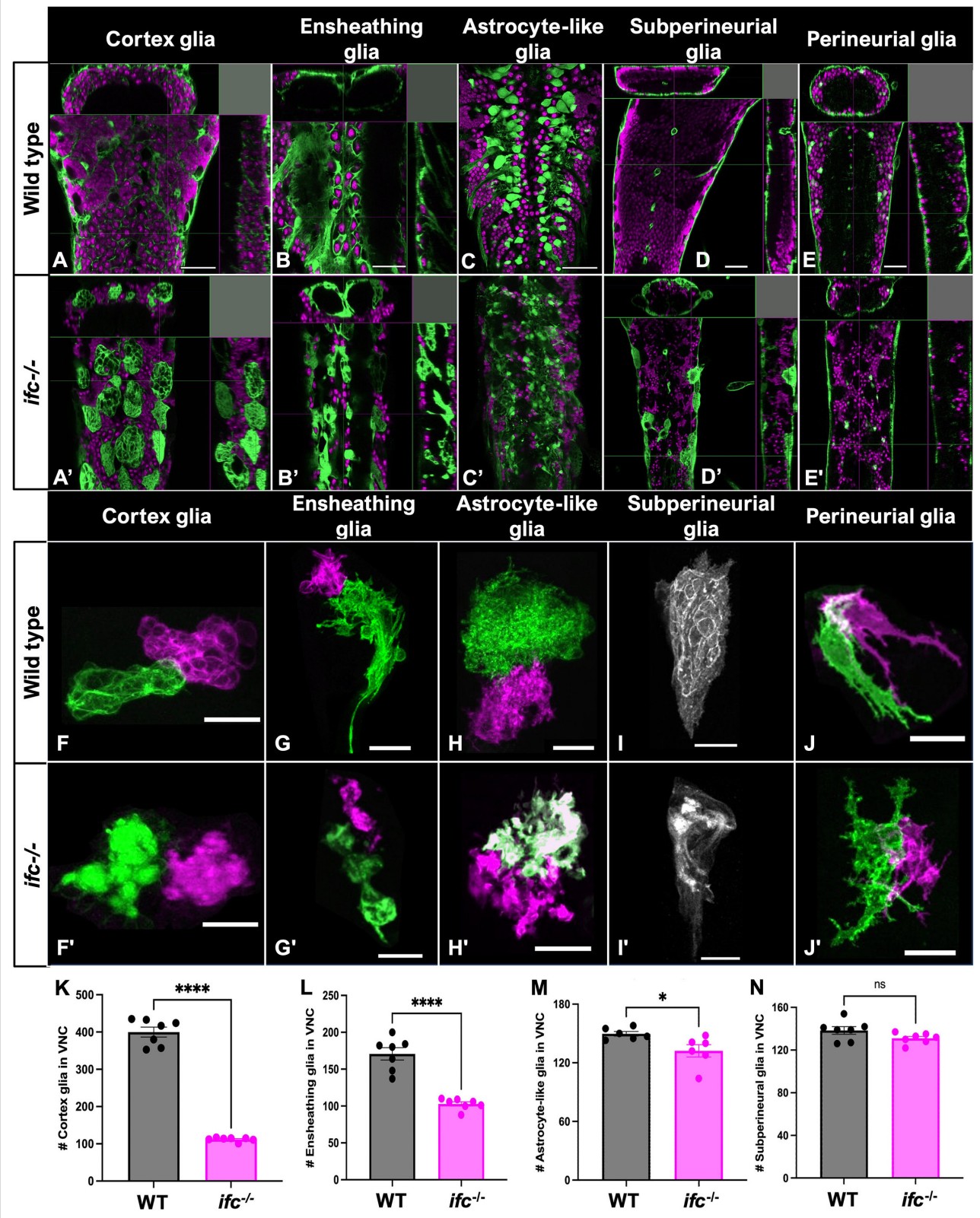

**Figure 2.** Loss of *ifc* disrupts glial morphology. (**A-E′**) High magnification ventral views and X–Z and Y–Z projections of the nerve cord of wild-type and *ifc*[−/−] late-third instar larvae labeled for ELAV (magenta) for neurons and Myr-GFP (green) for cell membranes of indicated glial subtype. Anterior is up; scale bar is 40 μm. (**F-J′**) High magnification views of individual glial cells of indicated glial subtype in the nerve cord of wild-type and *ifc*[−/−] larvae created by the MultiColor-FlpOut method (**Nern et al., 2015**). Anterior is up; scale bar is 20 μm. (**K–N**) Quantification of total number of indicated glial

*Figure 2 continued on next page*

*Figure 2 continued*

subtype in the nerve cord of wild-type and *ifc*⁻/⁻ late-third instar larvae (n = 7 for K, L, N; n = 6 for M). Statistics: *p < 0.05, ****p < 0.0001, and ns, not significant. The full genotype of flies shown in this figure can be found in *Supplementary file 1*.

The online version of this article includes the following figure supplement(s) for figure 2:

**Figure supplement 1.** Reduction of cortex glia number in *ifc*⁻/⁻ larvae results from increased apoptosis and reduced cell proliferation.

## *ifc* is predominately expressed in glia and localizes to the ER

Next, we tracked *ifc* expression in the CNS via RNA in situ hybridization and an *ifc-T2A-GAL4* transcriptional reporter. RNA in situ hybridization revealed that *ifc* is widely expressed in the CNS (*Figure 4A*), most obviously in the distinctive star-shaped astrocyte-like glia (*Figure 4B*), which are marked by Ebony expression (*Ziegler et al., 2013*). RNA in situ hybridization was not ideal for tracing *ifc* expression in other cells, likely due to signal diffusion. Thus, we paired the *ifc-T2A-GAL4* transcriptional reporter with a nuclear RFP (nRFP) transgene (*He et al., 2019*) and confirmed authenticity of the *ifc-T2A-GAL4* line by its strong expression in astrocyte-like glia (*Figure 4C*). Using this approach, we observed strong nRFP expression in all glial cells and modest nRFP expression in all neurons (*Figure 4D, E*, *Figure 4—figure supplement 1*), suggesting *ifc* is transcribed at higher levels in glial cells than neurons in the larval CNS.

Using a fosmid transgene that harbors a GFP-tagged version of *ifc* flanked by ~36 kb of its endogenous genomic region and molecular markers for ER, cis-Golgi, and trans-Golgi (*Sarov et al., 2016*; *Bergeron et al., 1994*; *Kikuma et al., 2017*; *Park et al., 2022*), we found that the Ifc-GFP colocalized strongly with the ER markers Calnexin 99A (CNX99A) and ESYT (*Figure 4F, F', G, G'*, *Figure 1—figure supplement 7*) and weakly with the cis-Golgi marker GOLGIN84 and the trans-Golgi marker

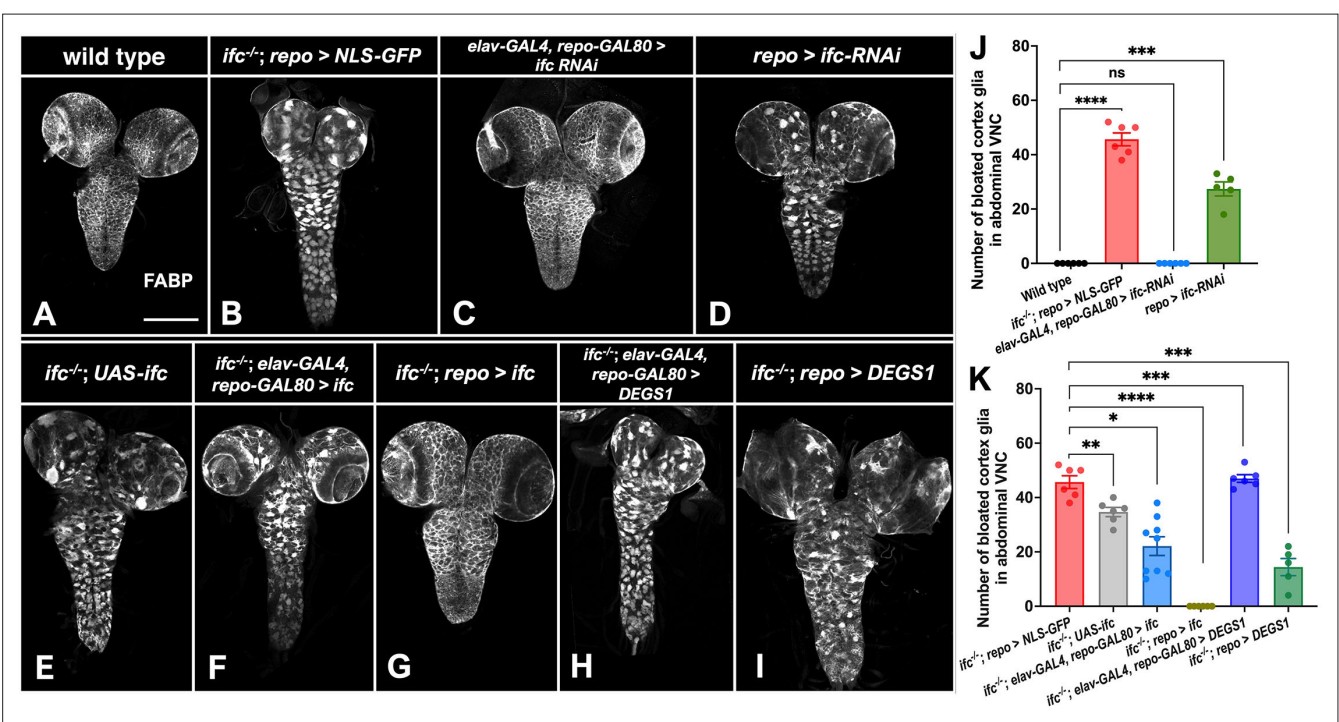

**Figure 3.** *ifc* acts in glia to regulate CNS structure and glial morphology. (**A–I**) Ventral views of photomontages of the CNS of late-third instar larvae labeled for fatty acid binding protein (FABP) (grayscale) to mark cortex glia in late-third instar larvae of indicated genotype. Neuronal-specific transgene expression was achieved by using *elav-GAL4* combined with *repo-GAL80*; glial-specific transgene expression was achieved by using *repo-GAL4*. (**J–K**) Quantification of the number of swollen cortex glia in the abdominal segments of the CNS of late-third instar larvae of the indicated genotype for the RNAi (**J**) and gene rescue assays (**K**). Statistics: *p < 0.05, **p < 0.01, ***p < 0.001, ****p < 0.0001, and ns, not significant. The full genotype of flies shown in this figure can be found in *Supplementary file 1*.

The online version of this article includes the following figure supplement(s) for figure 3:

**Figure supplement 1.** Glial-specific expression of *ifc* completely rescues the CNS phenotype of *ifc* mutant larvae.

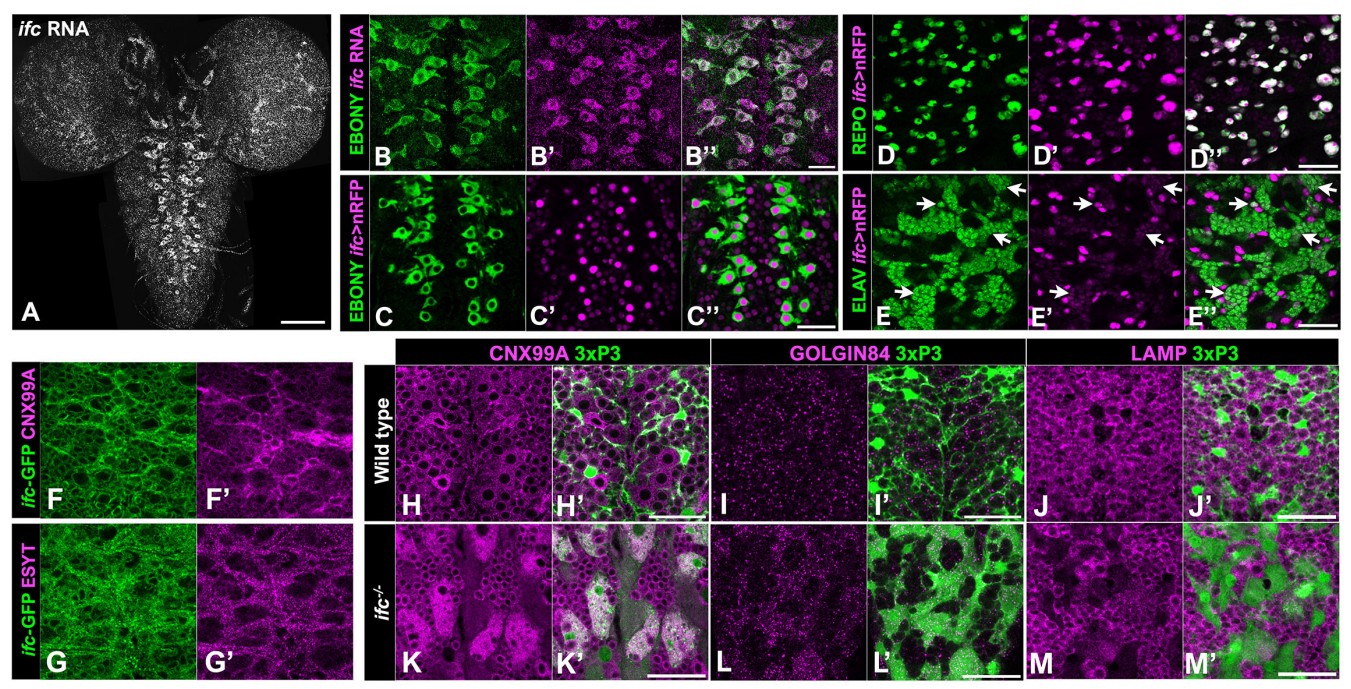

**Figure 4.** Loss of ifc drives ER expansion in cortex glia. Dorsal (**A, B–B”, C–C”**) and ventral (**D–D”, E–E”**) views of the CNS of late-third instar wild-type larvae labeled for *ifc* RNA (gray in A; magenta in B'), *ifc-GAL4>nRFP* (magenta; **C'–E'**), EBONY to mark astrocytes (green; **B, C**), REPO to mark glia (green; **D**), and ELAV to mark neurons (green; **E**). Panels D–D” and E–E” show surface and interior views, respectively, along the *Z*-axis on the ventral side of the nerve cord. Arrowheads in E–E” identify neurons with low-level *ifc-GAL4* expression. High magnification ventral views of thoracic segments in the CNS of wild-type late-third instar larvae labeled for GFP (green; **F, G**), CNX99A (magenta; **F'**), and ESYT (magenta; **G'**). (**H–M**) Late-third instar larvae of indicated genotype labeled for 3xP3-RFP (green; **H'–M'**), CNX99A (magenta; **H, K**), GOLGIN84 (magenta; **I, L**), and LAMP (magenta; **J, M**). Anterior is up; scale bar is 100 μm for panel A and 30 μm for panels B–M.

The online version of this article includes the following figure supplement(s) for figure 4:

**Figure supplement 1.** *ifc* is predominately expressed in glia and Ifc protein localizes to the Golgi apparatus, which appears to exhibit a mild expansion in the absence of *ifc* function in the larval CNS.

GOLGIN245 (*Figure 4—figure supplement 1*). Our results indicate that Ifc localizes primarily to the ER, aligning with the presumed site of de novo ceramide biosynthesis and prior work on *DEGS1* localization in cell lines (*Karsai et al., 2019*; *Kawano et al., 2006*).

## Loss of *ifc* drives ER expansion and lipid droplet loss in cortex glia

We next asked if loss of *ifc* function altered ER, Golgi, or lysosome morphology. Focusing on cortex glia, we observed a clear expansion of the ER marker CNX99A (*Figure 4H, H', K, K'*), a mild enrichment of the Golgi markers, GOLGIN84 and GOLGIN245, in diffuse 'clouds' (*Figure 4I, I', L, L', Figure 4—figure supplement 1*), and a reduction in expression of the lysosome marker LAMP (*Figure 4J, J', M, M'*) in *ifc*$^{-/-}$ larvae. The expansion of ER markers in *ifc* mutant larvae compelled us to obtain high-resolution views of organelle structure in cortex glia via transmission electron microscopy (TEM). In wild-type, cortex glia display a compact cytoplasm that surrounds a large nucleus (*Figure 5A, B*) and extend glial sheaths that fully enwrap adjacent neuronal cell bodies (black arrows; *Figure 5C, D*). In *ifc*$^{-/-}$ larvae, cortex glia display enlarged cell bodies with a maze-like pattern of internal membranes (solid white arrows; *Figure 5A', B', E, E'*) and fail to enwrap neurons (hollow white arrows; *Figure 5C', D'*). The internal membrane structures appear to assume an ER-like identity, as we observed significant overlap between the ER marker CNX99A and the membrane marker Myr-GFP when Myr-GFP was driven by a cortex glia-specific GAL4 line (*Figure 5F*). We observed similar yet milder effects on cell swelling and internal membrane accumulation in subperineurial and wrapping glia in abdominal nerves purple and pink shading, respectively (*Figure 5G, G', H, H'*), indicating that loss of *ifc* drives internal membrane accumulation in and swelling of multiple glial subtypes.

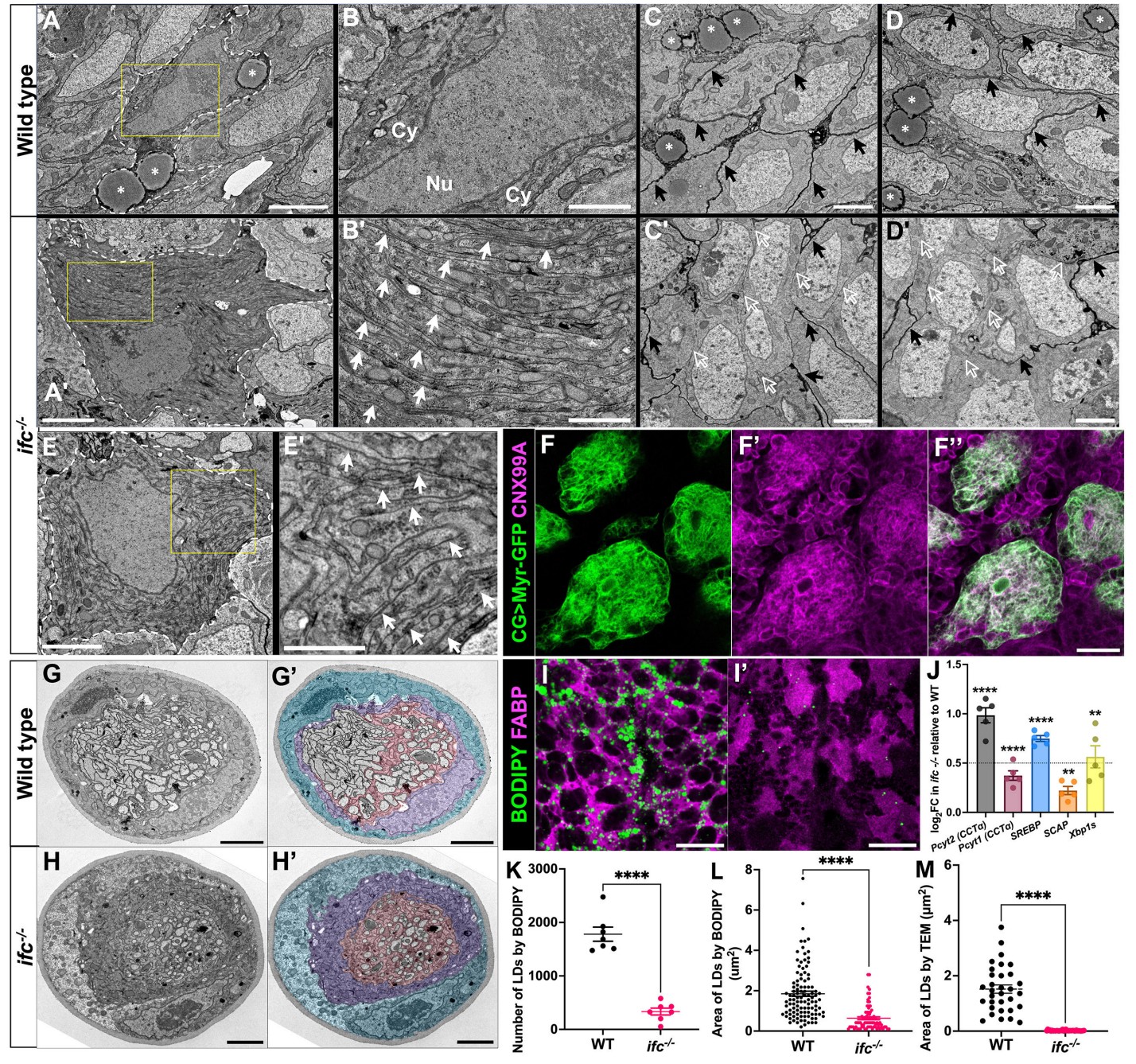

**Figure 5.** Loss of *ifc* leads to internal membrane accumulation and lipid droplet loss in cortex glia. (**A–E′**) Transmission electron microscopy (TEM) images of cortex glia cell body (**A, A′, B, B′**) and neuronal cell bodies (**C, C′, D, D′**) at low (**A, A′**) and high (**B, B′, C, C′, D, D′**) magnification in the nerve cord of wild-type (**A–D**) and *ifc*^−/− (**A′, D′, E, E′**) late-third instar larvae. (**A, A′**) Dotted lines demarcate cell boundary of cortex glia; yellow squares highlight regions magnified in B, B′, E′. Scale bar is 3 μm for A, A′ and 1 μm for B, B′. (**B, B′**) Cy denotes cytoplasm; Nu denotes nucleus. Solid white arrows highlight the layered internal membranes that occupy the cytoplasm of *ifc*^−/− cortex glia. (**C, C′, D, D′**) Black arrows highlight cortex glia membrane extensions that enwrap neuronal cell bodies; hollow white arrows denote the absence of cortex glia membrane extensions; white asterisk denotes lipid droplets. Scale bar is 2 μm. (**E, E′**) An additional example of membrane-filled cortex glia cell body in *ifc*^−/− larvae. Scale bar is 2 μm for E and 1 μm for E′. (**F**) Cortex glia in *ifc* mutant larvae labeled for Myr-GFP (green) to label membranes and CNX99A to label ER membranes. Scale bar is 30 μm. (**G, H**) Black and white and colored TEM cross-sections of peripheral nerves in wild-type and *ifc*^−/− late-third instar larvae. Blue marks perineurial glia; purple marks subperineurial glia; pink marks wrapping glia. Scale bar: 2 μm. High magnification ventral views of abdominal segments in the ventral nerve cord of wild-type (**I**) and *ifc* mutant (**I′**) third instar larvae labeled for BODIPY (green) to mark lipid droplets and fatty acid binding protein (FABP) (magenta) to label cortex glia. Anterior is up; scale bar is 30 μm. (**J**) Graph of log-fold change of transcription of five genes that promote membrane lipid

*Figure 5 continued on next page*

Figure 5 continued

synthesis in *ifc*<sup>−/−</sup> larvae relative to wild-type. A dotted line indicates a log$_2$ fold change of 0.5 in the treatment group compared to the control group. (**K–M**) Quantification of the number (**G**) and area of lipid droplets (**H, I**) in the dissected CNS of wild-type and *ifc*<sup>−/−</sup> larvae. Statistics: ****p < 0.0001, and ns, not significant.

The online version of this article includes the following figure supplement(s) for figure 5:

**Figure supplement 1.** ER chaperones are mostly downregulated, and genes involved in the Lands cycle are mostly upregulated in the CNS of *ifc*<sup>−/−</sup> mutant larvae.

TEM analysis also revealed a near complete depletion of lipid droplets in the CNS of *ifc*<sup>−/−</sup> larvae (compare **Figure 5A, C, D–A', C', D'**; lipid droplets marked by asterisk; **Figure 5M**), which we confirmed using BODIPY to mark neutral lipids in FABP-positive cortex glia (**Figure 5I–L**). In the CNS, lipid droplets form primarily in cortex glia (**Kis et al., 2015**) and are thought to contribute to membrane lipid synthesis through their catabolism into free fatty acids versus acting as an energy source in the brain (**Grabner et al., 2021**). Consistent with the possibility that increased membrane lipid synthesis drives lipid droplet reduction, RNA-seq assays of dissected nerve cords revealed that loss of *ifc* drove transcriptional upregulation of genes that promote membrane lipid biogenesis, such as *SREBP*, the conserved master regulator of lipid biosynthesis, *SCAP*, an activator of SREBP, and *Pcyt1/Pcyt2*, which promote phosphatidylcholine (PC) and phosphatidylethanolamine (PE) synthesis (**Eberlé et al., 2004**; **Osborne and Espenshade, 2009**; **Jacquemyn et al., 2017**). The spliced form of *Xbp-1* mRNA (*Xbp-1s*) (**Figure 5J**), which promotes membrane lipid synthesis required for ER biogenesis and is activated by the unfolded protein response (UPR) (**Sriburi et al., 2007**), is also upregulated in the CNS of *ifc* mutant larvae. Most ER chaperones, which are typically transcriptionally upregulated upon UPR activation (**Bernales et al., 2006**), were however downregulated (**Figure 5—figure supplement 1**), suggesting that in this case, misfolded protein is not the factor that triggers UPR activation and increased ER membrane biogenesis upon loss of *ifc*.

## Loss of *ifc* increases the saturation levels of triacylglycerols and membrane phospholipids

Lipid droplets are composed largely of triacylglycerols (TGs), **Jacquemyn et al., 2017** and we observed a five- and threefold drop in TG levels in the CNS and whole larvae of *ifc* mutant larvae relative to wild-type (**Figure 6A**). Our lipidomics analysis also revealed a shift of TGs toward higher saturation levels in the absence of *ifc* function (**Figure 6A–C**). Consistent with this, we observed transcriptional upregulation of most genes in the Lands cycle, which remodels phospholipids by replacing existing fatty acyl groups with new fatty acyl groups (**Figure 5—figure supplement 1**; **Moessinger et al., 2014**; **Moessinger et al., 2011**; **Harayama et al., 2014**; **O'Donnell, 2022**). As TG breakdown results in free fatty acids that can be used for membrane phospholipid synthesis, we asked if changes in TG levels and saturation were reflected in the levels or saturation of the membrane phospholipids PC, PE, and phosphatidylserine (PS). In the absence of the *ifc* function, PC and PE exhibited little change in quantity (**Figure 6D, G**), but the levels of the less abundant PS were increased threefold in the CNS of *ifc* mutant larvae relative to wild-type (compare **Figure 6J** to **Figure 6D, G**). All three phospholipids, however, displayed increased saturation levels: The relative levels of all PC, PE, and PS species were reduced, except for the most saturated form of each phospholipid – 18:1/18:1 – which is increased (**Figure 6E, F, H–L**). This increase was more pronounced in the CNS than whole larvae (**Figure 6F, I, L**), implying that loss of *ifc* function creates greater demand for lipid remodeling in the CNS.

## Reduction of dihydroceramide synthesis suppresses the *ifc* CNS phenotype

Following its synthesis, ceramide is transported by CERT from the ER to the Golgi, but CERT is less efficient at transporting dihydroceramide than ceramide (**Pan et al., 2023**). The expanded nature of the ER in *ifc* mutant larvae supports a model in which loss of *ifc* triggers excessive accumulation and retention of dihydroceramide in the ER due to CERT's inefficiency of transporting dihydroceramide to the Golgi, driving ER expansion and glial swelling. If this model is correct, reduction of dihydroceramide levels should suppress the *ifc* CNS phenotype. In agreement with this model, we found that the *schlank*<sup>G0365</sup> loss-of-function allele dominantly suppressed the enhanced RFP expression (**Figure 7M**)

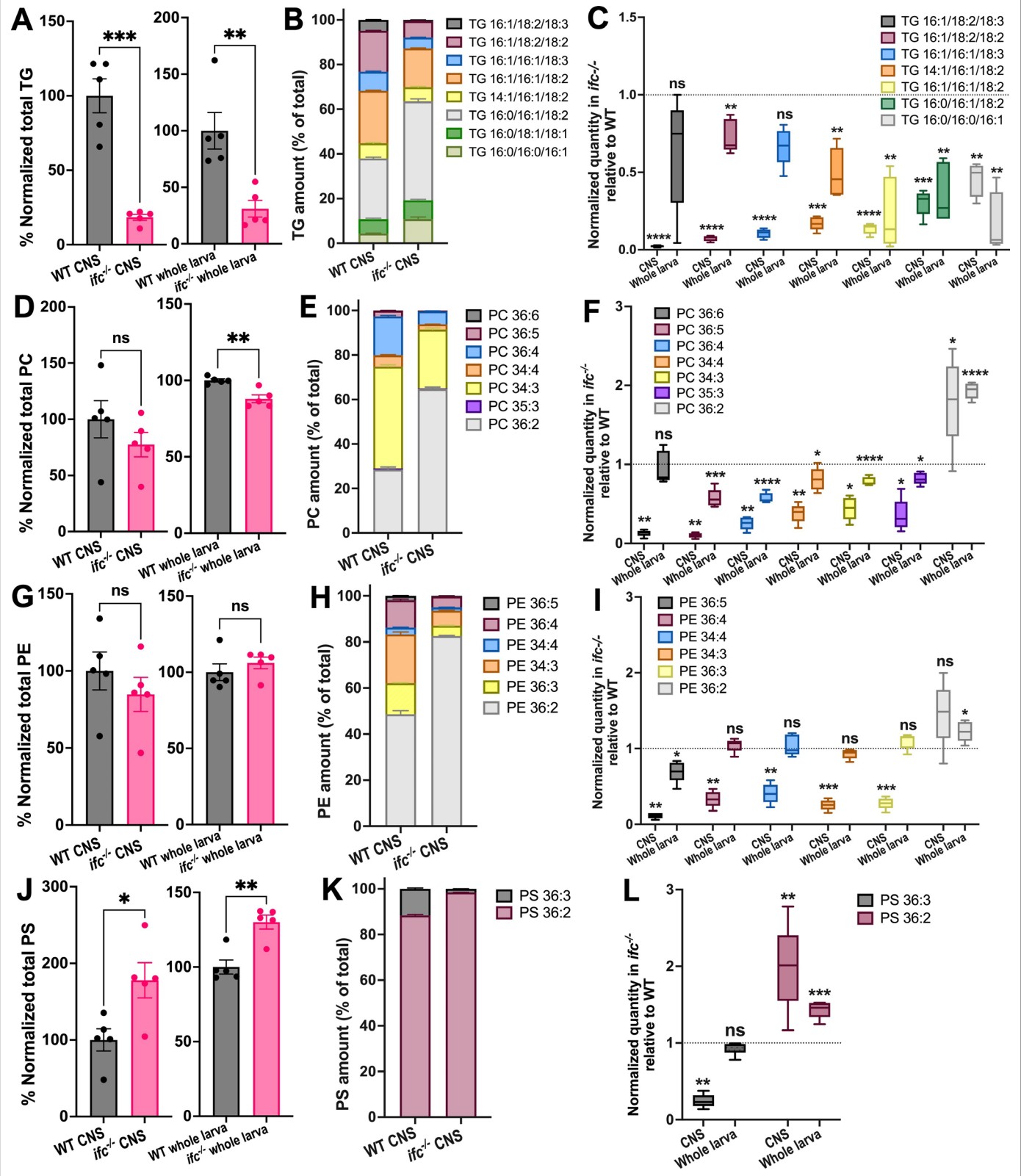

**Figure 6.** Phosphatidylcholine (PC), phosphatidylethanolamine (PE), phosphatidylserine (PS), and triacylglycerol (TG) exhibit higher saturation levels in a CNS-specific manner in *ifc* mutant late-third instar larvae. Quantification of total (**A**) and species-specific (**B, C**) TGs in whole larvae (**A, C**) and dissected CNS (A–C) of wild-type and *ifc*$^{-/-}$ larvae. Quantification of total (**D**) and species-specific (**E, F**) PCs in whole larvae (**D, F**) and dissected CNS (D–F) of wild-type and *ifc*$^{-/-}$ larvae. Quantification of total (**G**) and species-specific (**H, I**) PEs in whole larvae (**G, H**) and dissected CNS (G–I) of wild-type and *ifc*$^{-/-}$

*Figure 6 continued on next page*

*Figure 6 continued*

larvae. Quantification of total (**J**) and species-specific (**K, L**) PSs in the whole larvae (**J, L**) and dissected CNS (J–L) of wild-type and *ifc*⁻/⁻ larvae. Statistics: *p < 0.05, **p < 0.01, ***p < 0.001, ****p < 0.0001, and ns, not significant.

and CNS elongation phenotypes of *ifc* (*Figure 7N*). We also found that glial-specific depletion of *schlank* suppressed the internal membrane accumulation (*Figure 7A, B*), reduced lipid droplet size (*Figure 7C, D, O*), glial swelling (*Figure 7E, F*, *Figure 7—figure supplement 1*), enhanced RFP expression (*Figure 7M*), CNS elongation (*Figure 7N*), and reduced optic lobe (*Figure 1—figure supplement 5*) phenotypes observed in otherwise *ifc* mutant larvae. Our data support the model that inappropriate retention of dihydroceramide in the ER drives ER expansion and glial swelling and dysfunction.

## Glial-specific depletion of *ifc* function triggers neuronal cell death

Loss of *DEGS1/ifc* in human and flies has been shown to promote neurodegeneration and neuronal cell death (*Karsai et al., 2019*; *Pant et al., 2019*; *Jung et al., 2017*), but whether neuronal death results from an intrinsic defect in neurons or is induced by glial dysfunction remains unclear. By using Cleaved Caspase-3 as a marker of dying cells (*Yu et al., 2002*; *Fan and Bergmann, 2010*) and ELAV to track neurons, we tracked neuronal cell death in the brain and ventral nerve cord of wild-type and *ifc*⁻/⁻ mutant late-third instar larvae. In wild-type, little to no cell death was evident in the brain or nerve cord, and neurons appeared smoothly packed next to each other; in contrast, in *ifc*⁻/⁻ larvae, significant cell death was apparent in the brain and to a lesser degree in the nerve cord, with Caspase-3 staining often highlighting small perforations in the neuronal cell layer (*Figure 7G, H, P*, *Figure 1—figure supplement 7*). This perforated pattern was associated with and more expansive than Caspase-3 staining, indicating the perforations mark Caspase-positive dying neurons and Caspase-negative dead neurons. Glial-specific depletion of *schlank* function in otherwise *ifc*⁻/⁻ larvae suppressed the neuronal cell death phenotype (*Figure 7I, P*), supporting the model that loss of *ifc* function specifically in glia, rather than a specific requirement for *ifc* function in neurons, drives neuronal cell death. To directly test whether the *ifc* function in glia is required to guard against neuronal cell death, we used the GAL4-UAS system and RNAi-mediated gene interference to deplete *ifc* function specifically in glia (*repo-GAL4*) or in neurons (*elav-GAL4; repo-GAL80*) and found that glial-specific, but not neuronal-specific, depletion of *ifc* function drove significant neuronal cell death in the brain and to a greater extent the nerve cord, a phenotype that was enhanced upon removal of one copy of *ifc* (*Figure 7J, L, Q* and *Figure 7—figure supplement 2*). Loss of *ifc* function then triggers glial dysfunction, which in turn drives neuronal cell death.

## Discussion

Our work on *ifc* supports a model in which loss of *ifc/DEGS1* function drives glial dysfunction through the accumulation and inappropriate retention of dihydroceramide in the ER of glia with this proximal defect driving ER expansion, glial swelling, and subsequent neuronal cell death and neurodegeneration. A large increase in dihydroceramide would also impact ER membrane structure: ER membranes are typically loosely packed, thin, semi-fluid structures, with sphingolipids comprising just 3% of ER phospholipids (*Jacquemyn et al., 2017*). Sphingolipids in general and dihydroceramide in specific are highly saturated lipids, promote lipid order, tighter lipid packing, membrane rigidity, and thicker membranes (*O'Brien, 1965*; *Marcus and Popko, 2002*). Our observation of thick ER membranes in cortex glia in *ifc*⁻/⁻ larvae (*Figure 5*) aligns with increased dihydroceramide levels in the ER. In this context, we note that of many clinical observations made on an HLD-18 patient, one was widening of the ER in Schwann cells (*Karsai et al., 2019*), a finding that when combined with our work suggests that dihydroceramide accumulation in the ER is the proximal cause of HLD-18.

Although ER expansion represents the most proximal effect of loss of *ifc/DEGS1* function and dihydroceramide accumulation on cortex glia, other organelles and their functions are likely also disrupted, minimally as a consequence of ER disruption. For example, the apparent reduction of the lysosome marker, LAMP, in cortex glia of *ifc* mutant larvae correlates with increased RFP levels and the presence of bright RFP puncta or aggregates in this cell type, suggesting impaired lysosome function contributes to increased RFP perdurance and aggregation. Defects in the activity of multiple

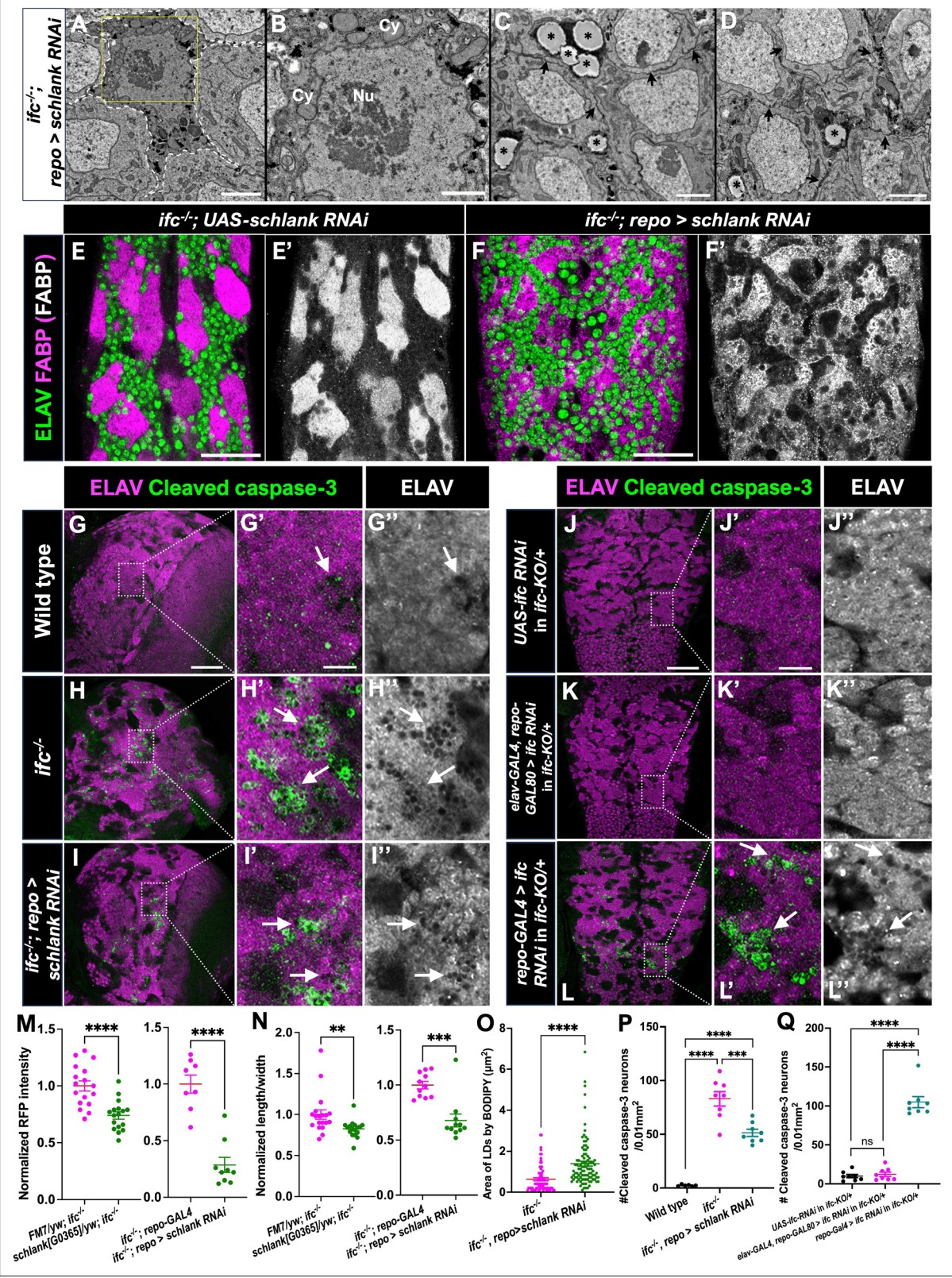

**Figure 7.** Glial-specific knockdown of *ifc* triggers neuronal cell death. Transmission electron microscopy (TEM) images of cortex glia cell body (**A, B**) and neuronal cell bodies (**C, D**) at low (**A**) and high (**B–D**) magnifications in the nerve cord of *ifc⁻/⁻; repo>schlank RNAi* late-third instar larvae. Dotted lines demarcate cell boundary of cortex glia; yellow squares highlight regions magnified in A. Scale bar: 3 µm for A, 1 µm for B, and 2 µm for C, D. (**B**) Cy denotes cytoplasm; Nu denotes nucleus. (**C, D**) Black asterisk denotes lipid droplets. Ventral views of abdominal sections of CNS of *ifc⁻/⁻; UAS-schlank*

*Figure 7 continued on next page*

*Figure 7 continued*

RNAi/+ larvae (**E**) and *ifc*$^{-/-}$; repoGAL4/UAS-schlank RNAi larvae (**F**) labeled for neurons (ELAV, green) and cortex glia (fatty acid binding protein [FABP], magenta/gray). Scale bar is 30 μm for E, F. Low (**G–L**) and high (**G'–L', G"–L"**) magnification views of the brain (**G–I**) and nerve cord (**J–L**) of late-third instar larvae of the indicated genotypes labeled for ELAV (magenta or grayscale) and Caspase-3 (green). Arrows indicate regions of high Caspase-3 signal and/or apparent neuronal cell death identified by perforations in the neuronal cell layer. Scale bar is 50 μm for panels G–L and 10 μm for panels G'–L". Quantification of CNS elongation (**M**) and 3xP3 RFP intensity (**N**) in *ifc* mutants alone, *ifc* mutants with one copy of *schlank[G0365]* loss-of-function allele, or *ifc* mutants in which *schlank* function is reduced via RNAi in glial cells. (**O**) Quantification of the area of lipid droplets in dissected CNS of *ifc* mutants and *ifc* mutants in which *schlank* function is reduced via RNAi in glial cells. Anterior is up in all panels. (**P, Q**) Quantification of Cleaved Caspase-3 neurons for panels G–I (**P**) and J–L (**Q**). Statistics: **p < 0.01, ***p < 0.001, ****p < 0.0001, and ns, not significant.

The online version of this article includes the following figure supplement(s) for figure 7:

**Figure supplement 1.** Glial-specific inhibition of *schlank* suppresses the *ifc* swollen cortex glia phenotype.

**Figure supplement 2.** Glial-specific, but not neuronal-specific, knockdown of *ifc* drives neuronal cell death.

**Figure supplement 3.** Subperineurial glial cell membranes encircle dying neurons in *ifc* mutant larvae.

organelles may collaborate to elicit the full phenotype manifested by loss of *ifc/DEGS1* function, resulting in glial dysfunction and subsequent neuronal cell death.

Increased dihydroceramide levels may contribute to a broader spectrum of neurodegenerative diseases than simply HLD-18. Recent work reveals that gain of function mutations in *SPTLC1* and *SPTLC2*, which encode components of the SPT complex that catalyzes the initial, rate-limiting step of de novo ceramide and sphingolipid biosynthesis (*Fan and Bergmann, 2010*), cause juvenile ALS via increased sphingolipid biosynthesis (*Lone et al., 2022*; *Lone et al., 2023*; *Syeda et al., 2024*; *Dohrn et al., 2024*; *Srivastava et al., 2023*). Of all sphingolipids, the relative levels of dihydroceramide were increased the most in patient plasma samples, suggesting that DEGS1 activity becomes limiting in the presence of enhanced SPT activity and that dihydroceramide accumulation contributes to juvenile ALS. Any mutations that increase metabolite flux through the ceramide pathway upstream of DEGS1 may then increase dihydroceramide levels, drive ER expansion and cell swelling, and lead to neurodegeneration, with disease severity predicated on the extent of excessive dihydroceramide accumulation. Model systems, like flies, can harness the power of genetic modifier screens to identify genes and pathways (potential therapeutic targets) that can be tweaked to ameliorate the effect of elevated dihydroceramide levels on neurodegeneration.

Our work appears to pinpoint glia as the cell type impacted by loss of *ifc/DEGS1* function, specifically glia that exhibit great demand for membrane biogenesis like cortex glia in the fly and oligodendrocytes or Schwann cells in mammals. In larvae, *ifc* is expressed at higher levels in glial cells than in neurons, and its genetic function is required at a greater level in glia than neurons to govern CNS development. Our unpublished work on other genes in the ceramide metabolic pathway reveals similar glial-centric expression patterns in the larval CNS to that observed for *ifc*, suggesting they too function primarily in glia rather than neurons at this stage. In support of this model, a study on CPES, which converts ceramide into CPE, the fly analog of sphingomyelin, revealed that CPES is required in cortex glia to promote their morphology and homeostasis and to protect flies from photosensitive epilepsy (*Kunduri et al., 2018*). Similarly, ORMDL, a dedicated negative regulator of the SPT complex, is required in oligodendrocytes to maintain proper myelination in mice (*Clarke et al., 2019*). Given that many glial cell types are enriched in sphingolipids and exhibit a great demand for new membrane biogenesis during phases of rapid neurogenesis and axonogenesis, we believe that glial cells, such as cortex glia, oligodendrocytes, and Schwann cells, rather than neurons manifest a greater need for *ifc/DEGS1* function and ceramide synthesis during developmental stages marked by significant nervous system growth.

Will this glial-centric model for *ifc/DEGS1* function, and more generally ceramide synthesis, hold true in the adult when neurogenesis is largely complete and the demand for new membrane synthesis in glia dissipates? Recent work in the adult fly eye suggests it may not (*Wang et al., 2022*). argued that the GlcT enzyme, which converts ceramide to glucosylceramide, is expressed at much higher levels in neurons than glia, and that glucosylceramide is then transported from neurons to glial cells for its degradation, suggesting cell-type-specific compartmentalization of sphingolipid synthesis in neurons and its degradation in glia in the adult. In the future, it will be exciting to uncover whether genes of the sphingolipid metabolic pathway alter their cell-type-specific requirements as a function of developmental stage.

We note that cortex glia are the major phagocytic cell of the CNS and phagocytose neurons targeted for apoptosis as part of the normal developmental process (*Freeman and Doherty, 2006*; *Coutinho-Budd and Freeman, 2013*; *Yildirim et al., 2019*; *Corty and Coutinho-Budd, 2023*). Thus, while we favor the model that *ifc* triggers neuronal cell death due to glial dysfunction, it is also possible that increased detection of dying neurons arises due at least in part to a decreased ability of cortex glia to clear dying neurons from the CNS. At present, the large number of neurons that undergo developmentally programmed cell death combined with the significant disruption to brain and ventral nerve cord morphology caused by loss of *ifc* function renders this question difficult to address. Additional evidence does, however, support the idea that loss of *ifc* function drives excess neuronal cell death: Clonal analysis in the fly eye reveals that loss of *ifc* drives photoreceptor neuron degeneration (*Jung et al., 2017*), indicating that loss of *ifc* function drives neuronal cell death; cortex-glia-specific depletion of *CPES*, which acts downstream of *ifc*, disrupts neuronal function and induces photosensitive epilepsy in flies (*Kunduri et al., 2018*), indicating that genes in the ceramide pathway can act non-autonomously in glia to regulate neuronal function; recent genetic studies reveal that other glial cells can compensate for impaired cortex glial cell function by phagocytosing dying neurons (*Beachum et al., 2024*), and we observe that the cell membranes of subperineurial glia enwrap dying neurons in *ifc* mutant larvae (*Figure 7—figure supplement 3*), consistent with similar compensation occurring in this background, and in humans, loss-of-function mutations in *DEGS1* cause neurodegeneration (*Dolgin et al., 2019*; *Karsai et al., 2019*; *Pant et al., 2019*). Clearly, future work is required to address this question for *ifc/DEGS1* and perhaps other members of the ceramide biogenesis pathway.

Altered glial function may not only derive from dihydroceramide buildup in the ER, but also from altered cell membrane structure due to the replacement of ceramide and its derivatives, such as GlcCer and CPE, with the cognate forms of dihydroceramide. Relative to dihydroceramide species, the 4–5 *trans* carbon–carbon double bond in the sphingoid backbone of ceramide-containing sphingolipids enables them to form more stable hydrogen bonds with water molecules and facilitates their ability to associate with lipids of different saturation levels (*Li et al., 2002*). A high dihydroceramide to ceramide ratio has been shown to form rigid gel-like domains within model membranes and to destabilize biological membranes by promoting their permeabilization (*Vieira et al., 2010*; *Hernández-Tiedra et al., 2016*). As even minor alterations to membrane properties can disrupt glial morphology (*Aggarwal et al., 2011*), such alterations in membrane rigidity and stability may underlie the failure of cortex glia to enwrap adjacent neurons. The observed increase in saturation of PE, PC, and PS in the CNS of *ifc* mutant larvae may reflect a compensatory response employed by cells to stabilize cell membranes and guard cell integrity when challenged with elevated levels of dihydroceramide.

The expansion of ER membranes coupled with loss of lipid droplets in *ifc* mutant larvae suggests that the apparent demand for increased membrane phospholipid synthesis may drive lipid droplet depletion, as lipid droplet catabolism can release free fatty acids to serve as substrates for lipid synthesis. At some point, the depletion of lipid droplets, and perhaps free fatty acids as well, would be expected to exhaust the ability of cortex glia to produce additional membrane phospholipids required for fully enwrapping neuronal cell bodies. Under wild-type conditions, many lipid droplets are present in cortex glia during the rapid phase of neurogenesis that occurs in larvae. During this phase, lipid droplets likely support the ability of cortex glia to generate large quantities of membrane lipids to drive membrane growth needed to ensheath newly born neurons. Supporting this idea, lipid droplets disappear in the adult *Drosophila* CNS when neurogenesis is complete and cortex glia remodeling stops (*Kis et al., 2015*). We speculate that lipid droplet loss in *ifc* mutant larvae contributes to the inability of cortex glia to enwrap neuronal cell bodies. Prior work on lipid droplets in flies has focused on stress-induced lipid droplets generated in glia and their protective or deleterious roles in the nervous system (*Bailey et al., 2015*; *Liu et al., 2015*; *Liu et al., 2017*). Work in mice and humans has found that more lipid droplets are often associated with the pathogenesis of neurodegenerative diseases (*Marshall et al., 2014*; *Farmer et al., 2020*; *Marschallinger et al., 2020*), but our work correlates lipid droplet loss with CNS defects. In the future, it will be important to determine how lipid droplets impact nervous system development and disease.

## Methods

### Methods details

#### Fly husbandry

Flies were raised on standard molasses-based food at 25°C. Unless otherwise noted, wild-type is an otherwise wild-type stock harboring the *M{3xP3-RFP.attP}ZH-51D* insert in an isogenic second chromosome.

#### Mutagenesis

A standard autosomal recessive forward genetic screen was carried out using 25–30 μm EMS to mutagenize a *M{3xP3-RFP.attP}ZH-51D* isogenic second chromosome. Homozygous mutant third instar larvae were visually screened under a standard fluorescent microscope for defects in CNS morphology. A detailed description of the screen, all identified genes, and associated whole-genome sequences is described elsewhere (*Lacin et al., 2024*).

#### Creation of recombinant lines and identification of second site mutations in *ifc-KO* chromosome

Standard genetic methods were used to generate fly strains that contained specific combinations of GAL4 and UAS-linked transgenes in the *ifc^js3^* or *ifc-KO* background. During this process, we uncovered that the *ifc-KO* deletion could be unlinked from at least one second chromosomal mutation that caused early larval lethality, resulting in homozygous *ifc-KO* flies that survived to late L3 to early pupa. A subsequent EMS-based F2 lethal non-complementation screen using the *M{3xP3-RFP.attP} ZH-51D* isogenic second chromosome as target chromosome and screening against the *ifc-KO* chromosome identified multiple mutations in four complementation groups that led to an early larval lethal phenotype. Whole-genome sequencing identified these genes as *Med15*, *lwr*, *Nle*, and *Sf3b1*; all four genes reside within ~90 kb of each other in chromosomal bands 21D1–21E2 near the telomere in chromosome 2L, identifying the site of the associated lethal mutations and suggesting the actual lesion may be a small deletion that removes these genes. The following alleles of these genes are available at Bloomington Stock Center: *Med15^js1^* (Q175*), *Med15^js2^* (Q398*), *Med15^js3^* (C655Y), *lwr^js1^* (I4N), *lwr^js2^* (E12K), *Nle^js1^* (W146R), *Nle^js2^* (L125P), *Nle^js3^* (Q242*), *Sf3b1^js1^* (R1160*), *Sf3b1^js2^* (Q1264*), *Sf3b1^js3^* (570 bp deletion at the following coordinates chr2L571720–572290; this deletion removes amino acids 1031 through 1222 and introduces a frameshift into the reading frame).

#### Gene rescue and in vivo RNAi phenocopy assays

To restrict UAS-linked transgene expression specifically to glia, we used the *repoGAL4* driver line. To restrict UAS-linked transgene expression specifically to neurons, we paired the *elavGAL4* driver lines, which activate transgene expression strongly in all neurons and moderately in glia, with *repoGAL80*, which blocks GAL4-dependent activation in glia. We used *P{VSH330794}* (VDRC 330794) for the RNAi experiments and *UAS-ifc* (*Bischof et al., 2013*) and *UAS-DEGS1* (BDSC 79200) for the gene rescue assays. All gene rescue experiments were performed in the *ifc^js3^*/*ifc-KO* background with the UAS-transgene placed into the *ifc^js3^* background and the GAL4 drivers into the *ifc-KO* background.

The GAL4-UAS method was also used to assess the phenotype of each glial subtype. In combination with the *UAS-Myr-GFP* transgene, which labels cell membranes, we used the following GAL4 lines to trace the morphology of each glial subtype in wild-type and *ifc^−/−^* larvae: *GMR85G01-GAL4* (perineurial glia; BDSC 40436), *GMR54C07-GAL4* (subperineurial glia; BDSC 50472), *GMR54H02-GAL4* (cortex glia; BDSC 45784), *GMR56F03-GAL4* (ensheathing glia; BDSC 77469 and 39157), and *GMR86E01-GAL4* (astrocyte-like glia; BDSC45914). The *UAS-Myr-GFP* transgene was placed into the *ifc^js3^* background; each glial GAL4 line was placed into the *ifc-KO* background.

The GAL4-UAS method was used to assess the ability of *UAS-ifc* and *UAS-DEGS1* transgenes to rescue the lethality of otherwise *ifc* mutant larvae. *ifc-KO/CyO Tb; repo-GAL4/TM6B Tb* males were crossed to each of the following lines: *ifc^js3^/CyO Tb; UAS-ifc; ifc^js3^/CyO Tb; UAS-DEGS1; ifc^js3^/CyO Tb.* All adult flies were sorted into Curly and non-Curly and then counted; all non-Curly flies lacked the *TM6B Tb* balancer indicating they carried *repo-GAL4* and thus were likely rescued to viability by glial expression of *ifc* or *DEGS1*. Standard Mendelian ratios were then used to predict the expected number of *ifc* mutant flies if all survived to adulthood. The total number of observed adult *ifc* mutant

flies, identified by lack of Curly wings, was then divided by this number to obtain the percentage of *ifc* mutant flies that survived to adulthood. 2452 total flies were assayed for the *UAS-ifc* cross, 1303 for the *UAS-DEGS1* cross, and 1030 for the control cross.

## DNA sequencing

Genomic DNA was obtained from wild-type larvae or larvae homozygous for each relevant mutant line and provided to GTAC (Washington University) or GENEWIZ for next-generation or Sanger sequencing.

RNA in situ hybridization *ifc* RNA probes for in situ hybridization chain reaction (HCR) were designed and made by Molecular Instruments (HCR RNA-FISH v3.0) (*Choi et al., 2018*). Wild-type CNS was harvested and fixed in 2% paraformaldehyde at late L3. The fixed CNS underwent gradual dehydration and rehydration, followed by standard hybridization and amplification steps of the HCR protocol (*Choi et al., 2018*). For double labeling with antibody, the post-HCR labeled CNS was briefly fixed for 30 min prior to standard antibody labeling protocol to identify specific CNS cell type(s) with highly localized *ifc* RNAs (*Lacin et al., 2019*; *Duckhorn et al., 2022*).

## MultiColor FlpOut labeling of glial subtypes

For glial labeling in the control background, the *MCFO1* line was crossed to *GMR-GAL4* driver lines for each glial subtype (see Key Resources Table; *Nern et al., 2015*). For glial labeling in the *ifc*$^{-/-}$ background, the *MCFO1* line was placed in the *ifc*$^{JS3}$ background, each of the five glial-specific GMR-GAL4 lines was placed individually into the *ifc-KO* background, and then the MCFO, *ifc*$^{JS3}$ line was crossed to each glial-specific GAL4, *ifc-KO* line. Flies were allowed to lay eggs for 24 hr at 25°C, and progeny were raised at 25°C for 4 days prior to heat-activated labeling. On day 4 after egg-laying, F1 larvae were incubated in a 37°C water bath for 5 min. When wild-type or *ifc*$^{-/-}$ mutant larvae reached late L3, which was days 5 and 6 for control and days 9 and 10 for *ifc*$^{-/-}$ mutants, the CNS was dissected, fixed, stained, and then analyzed under a Zeiss LSM 700 Confocal Microscope for the presence of clones, using Zen software.

## Antibody generation

YenZym (CA, USA) was used as a commercial source to generate affinity purified antibodies against a synthetic peptide that corresponded to amino acids 85–100 (TLDGNKLTQEQKGDKP) of FABP isoform B. Briefly, the peptide was conjugated to KLH, used as an immunogen in rabbits to generate a peptide-specific antibody response, and antibodies specific to the peptide were affinity purified. The affinity-purified antibodies were confirmed to label cortex glia specifically based on comparison of antibody staining relative to Myr-GFP when a UAS-Myr-GFP was driven under control of the cortex-specific glial GAL4 driver GMR-54H02 (*Figure 1—figure supplement 3*). The FABP antibodies are used at 1:500–1:1000.

## Immunofluorescence and lipid droplet staining

Gene expression analysis was performed essentially as described in *Patel, 1994*. Briefly, the larval CNS was dissected in PBS, fixed in 2.5% paraformaldehyde for 55 min, and washed in PTx 1× PBS, 0.1% Triton X-100. The fixed CNS was incubated in primary antibody with gentle rocking overnight at 4°C. Secondary antibody staining was conducted for at least 2 hr to overnight at room temperature. All samples were washed in PTx at least five times and rocked for an hour before and after secondary antibody staining. A detailed list of the primary and secondary antibodies is available in the Key Resources Table. Dissected CNS were mounted either in PTx or dehydrated through an ethanol series and cleared in xylenes prior to mounting in DPX mountant (*Truman et al., 2004*). All imaging was performed on a Zeiss LSM-700 Confocal Microscope, using Zen software.

For lipid droplet staining, the fixed CNS was incubated for 30 min at room temperature at 1:200 dilution of 1 mg/ml BODIPY 493/503 (Invitrogen: D3922). It was then rinsed thoroughly in PBS and immediately mounted for imaging on a Zeiss LSM-700 Confocal Microscope, using Zen software.

## TEM imaging

For TEM, samples were immersion fixed overnight at 4°C in a solution containing 2% paraformaldehyde and 2.5% glutaraldehyde in 0.15 M cacodylate buffer with 2 mM $CaCl_2$, pH 7.4. Samples were then rinsed in cacodylate buffer three times for 10 min each and subjected to a secondary fixation for 1 hr in 2% osmium tetroxide/1.5% potassium ferrocyanide in cacodylate buffer. Following this, samples were rinsed in ultrapure water three times for 10 min each and stained overnight in an aqueous solution of 1% uranyl acetate at 4°C. After staining was complete, samples were washed in ultrapure water 3 times for 10 min each, dehydrated in a graded acetone series (50%, 70%, 90%, 100% ×4) for 15 min in each step, and infiltrated with microwave assistance (Pelco BioWave Pro, Redding, CA) into Spurr's resin. Samples were then cured in an oven at 60°C for 72 hr and post-curing, 70 nm thin sections were cut from the resin block, post-stained with uranyl acetate and Sato's lead and imaged on a Transmission Electron Microscope (JEOL JEM-1400 Plus, Tokyo, Japan) operating at 120 keV.

## Lipidomics

Untargeted lipidomics analysis was conducted on whole larva and dissected CNS of wild-type and $ifc^{-/-}$ mutants at the late-third instar stage. Five replicates were prepared for each set of experiments. For whole larvae, at least 15 larvae of each genotype were used for each replicate. For the dissected CNS, at least 50 wild-type and 60 $ifc^{-/-}$ CNS were used per replicate. Immediately following collection or dissection, larvae and the dissected CNS were flash frozen in liquid nitrogen and placed at –80°C.

Lipids were extracted from frozen whole larvae and dissected larval CNS samples by using an Omni Bead Ruptor Elite Homogenizer using acetonitrile:methanol:water (2:2:1; 40 µl/mg tissue). Two ultrahigh-performance LC (UHPLC)/MS systems were used in this work: a Thermo Vanquish Flex UHPLC system with a Thermo Scientific Orbitrap ID-X and an Agilent 1290 Infinity II UPLC system with an Agilent 6545 QTOF as described previously (*Cho et al., 2021*). Lipids were separated on a Waters Acquity HSS T3 column (150 × 2.1 mm, 1.8 mm). The mobile-phase solvents were composed of A: 0.1% formic acid, 10 mM ammonium formate, 2.5 µM medronic acid in 60:40 acetonitrile:water; and B = 0.1% formic acid, 10 mM ammonium formate in 90:10 2-propanol:acetonitrile. The column compartment was maintained at 60°C. The following linear gradient was applied at a flow rate of 0.25 ml min$^{-1}$: 0–2 min, 30% B; 17 min, 75% B; 20 min, 85% B; 23–26 min, 100% B. The injection volume was 4 µl for all lipids analysis. Data were acquired in positive ion.

LC/MS data were processed and analyzed with the open-source Skyline software (*Adams et al., 2020*). Lipid MS/MS data were annotated with Agilent Lipid Annotator software.

## RNA sequencing and analysis

To determine the CNS-specific transcriptional changes upon loss of *ifc*, RNA-seq was conducted on five replicates of dissected CNS tissue derived from wild-type and $ifc^{-/-}$ mutant late-third instar larvae. For each replicate, roughly 30–35 dissected CNS of wild-type or $ifc^{-/-}$ larvae were used. Invitrogen RNAqueous-Micro Total RNA Isolation Kit (AM1931) was used to extract RNA. Agilent 4200 TapeStation system was used for RNA quality control. All RNA samples were then provided to the Genome Technology Access Center in the McDonnell Genome Institute at Washington University for next-generation sequencing.

Samples were prepared according to library kit manufacturer's protocol, indexed, pooled, and sequenced on an Illumina NovoSeq. Basecalls and demultiplexing were performed with Illumina's bcl2fastq software with a maximum of one mismatch in the indexing read. RNA-seq reads were then aligned to the Ensembl release 76 primary assembly with STAR version 2.5.1a (*Dobin et al., 2013*) Gene counts were derived from the number of uniquely aligned unambiguous reads by Subread:-featureCount version 1.4.6-p5 (*Liao et al., 2014*). Isoform expression of known Ensembl transcripts was estimated with Salmon version 0.8.2 (*Patro et al., 2017*). To find the most critical genes, the raw counts were variance stabilized with the R/Bioconductor package DESeq2 (*Love et al., 2014*) and were then analyzed via weighted gene correlation network analysis with the R/Bioconductor package WGCNA (*Langfelder and Horvath, 2008*).

## Statistics

All data are presented as mean ± SEM. Statistical significance between groups was determined using Student's *t*-test or one-way ANOVA with multiple comparisons, and with varying levels of significance

assessed as *p < 0.05, **p < 0.01, ***p < 0.001, ****p < 0.0001, and ns, not significant. All experiments were done with a minimum of five biological replicates.

## Material availability statement

All newly created reagents or strains are freely available by contacting the corresponding author and/or through public stock centers at which many of the strains were deposited.

## Acknowledgements

We thank the Iowa Developmental Studies Hybridoma Bank for antibodies, and the Bloomington Stock Center, Vienna *Drosophila* Research Center, FlyORF, and the National Institutes of Genetics stock center in Japan for countless fly lines. We thank Drs. Christian Klambt, Chih-Chiang Chan, and Dion Dickman for reagents. We thank Dr. Tristan Qingyuan Li for comments on the manuscript. We thank the Genome Technology Access Center at Washington University for next-generation sequencing and analysis of RNA-seq samples. We thank Dr. Sanja Sviben, Gregory Strout, and John Wulf II for assistance in TEM studies conducted at the Washington University Center for Cellular Imaging, which is supported in part by Washington University School of Medicine, The Children's Discovery Institute of Washington University and St. Louis Children's Hospital (CDI-CORE-2015-505 and CDI-CORE-2019-813), the Foundation for Barnes-Jewish Hospital (3770), and the Washington University Diabetes Research Center (NIH P30 DK020579).

## Additional information

### Competing interests

Gary Patti: Collaborative research agreement with Agilent Technologies and Thermo Fisher. Chief Scientific Officer of Panome Bio. The other authors declare that no competing interests exist.

### Funding

| Funder | Grant reference number | Author |
| --- | --- | --- |
| National Institute of Neurological Disorders and Stroke | NS036570 | James B Skeath |
| National Institute of Neurological Disorders and Stroke | NS122903 | Haluk Lacin |
| National Institute of Environmental Health Sciences | ES2028365 | Gary Patti |

The funders had no role in study design, data collection, and interpretation, or the decision to submit the work for publication.

### Author contributions

Yuqing Zhu, Haluk Lacin, Conceptualization, Data curation, Formal analysis, Validation, Investigation, Visualization, Methodology, Writing – original draft, Writing – review and editing; Kevin Cho, Data curation, Formal analysis, Investigation, Methodology, Writing – review and editing; Yi Zhu, Data curation, Formal analysis, Validation, Investigation, Methodology, Writing – review and editing; Jose T DiPaola, Formal analysis, Validation, Investigation, Visualization, Methodology; Beth A Wilson, Data curation, Formal analysis, Validation, Investigation, Methodology; Gary Patti, Data curation, Formal analysis, Funding acquisition, Investigation, Methodology, Project administration; James B Skeath, Conceptualization, Resources, Data curation, Formal analysis, Supervision, Funding acquisition, Validation, Investigation, Visualization, Methodology, Writing – original draft, Project administration, Writing – review and editing

### Author ORCIDs

Yuqing Zhu https://orcid.org/0000-0001-8693-4741

Haluk Lacin [ID] https://orcid.org/0000-0003-2468-9618
Gary Patti [ID] https://orcid.org/0000-0002-3748-6193
James B Skeath [ID] https://orcid.org/0000-0003-1179-4857

Reviewer #1 (Public review): https://doi.org/10.7554/eLife.99344.3.sa1
Reviewer #2 (Public review): https://doi.org/10.7554/eLife.99344.3.sa2
Reviewer #3 (Public review): https://doi.org/10.7554/eLife.99344.3.sa3
Author response https://doi.org/10.7554/eLife.99344.3.sa4

## Additional files

### Supplementary files
Supplementary file 1. Full genotype of flies shown in *Figures 2 and 3*.

MDAR checklist

### Data availability
Sequencing data have been deposited in Geo under accessions code: GSE263308. Untargeted lipid-omics data have been deposited in the Metabolics Workbench under project ID - PR001967 and study ID - ST003162.

The following datasets were generated:

| Author(s) | Year | Dataset title | Dataset URL | Database and Identifier |
|---|---|---|---|---|
| Skeath JB, Zhu Y, Zhu Y, Lacin H, Wilson BA | 2024 | Loss of dihydroceramide desaturase drives neurodegeneration by disrupting endoplasmic reticulum and lipid droplet homeostasis in glial cells | https://www.ncbi.nlm.nih.gov/geo/query/acc.cgi?acc=GSE263308 | NCBI Gene Expression Omnibus, GSE263308 |
| Cho K, Patti GJ, Skeath JB | 2024 | Loss of dihydroceramide desaturase drives neurodegeneration by disrupting endoplasmic reticulum and lipid droplet homeostasis in glial cells | http://dx.doi.org/10.21228/M8G72D | Metabolics Workbench, 10.21228/M8G72D |

The following previously published dataset was used:

| Author(s) | Year | Dataset title | Dataset URL | Database and Identifier |
|---|---|---|---|---|
| Lacin H, Zhu Y, DiPaola JT, Wilson BA, Zhu Y, Skeath JB | 2024 | A Genetic Screen for regulators of CNS morphology in *Drosophila* | https://www.ncbi.nlm.nih.gov/bioproject/?term=PRJNA1128589 | NCBI BioProject, PRJNA1128589 |

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

# Appendix 1

## Appendix 1—key resources table

| Reagent type (species) or resource | Designation | Source or reference | Identifiers | Additional information |
|---|---|---|---|---|
| Gene (*Drosophila melanogaster*) | Infertile crescent (ifc) | GenBank | FLYB: FBgn-0001941 | |
| Genetic reagent (*D. melanogaster*) | y[1] M{RFP [3xP3.PB] GFP [E.3xP3=vas-int.Dm]ZH-2A w[*]; M{3xP3RFP. attP}ZH-51D | Bloomington *Drosophila* Stock Center | RRID:BDSC24483 | Wild-type. Control for 3xP3 background. |
| Genetic reagent (*D. melanogaster*) | Gpdh1[nSP4] ifc[1] wg[spd-fg] pr[1]/CyO, cl[4] | Bloomington *Drosophila* Stock Center | RRID:BDSC4963 | |
| Genetic reagent (*D. melanogaster*) | w[1118]; Df(2L)ED334/SM6a | Bloomington *Drosophila* Stock Center | RRID:BDSC9343 | Deficiency line that uncovers ifc |
| Genetic reagent (*D. melanogaster*) | w[1118]; Df(2L)BSC184/CyO | Bloomington *Drosophila* Stock Center | RRID:BDSC9612 | Deficiency line that uncovers ifc |
| Genetic reagent (*D. melanogaster*) | w[1118]; Df(2L)BSC353/CyO | Bloomington *Drosophila* Stock Center | RRID:BDSC24377 | Deficiency line that uncovers ifc |
| Genetic reagent (*D. melanogaster*) | y[1] w[*]; TI{GFP[3xP3.cLa]=CRIMIC.TG4.2} ifc[CR70115-TG4.2]/SM6a | Bloomington *Drosophila* Stock Center | RRID:BDSC92710 | |
| Genetic reagent (*D. melanogaster*) | v[1]; Kr[If-1]/CyO; P{y[+t7.7] v[+t1.8]=UAS-Trans-Timer.v+}attP2 | Bloomington *Drosophila* Stock Center | RRID:BDSC93411 | |
| Genetic reagent (*D. melanogaster*) | y[1] w[*]; PBac {y[+mDint2] w[+mC]=UAS-hDEGS1.HA}VK00033 | Bloomington *Drosophila* Stock Center | RRID:BDSC79200 | UAS line for human DEGS1 |
| Genetic reagent (*D. melanogaster*) | w[1118] P{y[+t7.7] w[+mC]=hs-FLP G5.PEST}attP3;PBac{y[+mDint2] w[+mC]=10xUAS(FRT.stop) myr::smGdP-HA} VK0000 5 P{y[+t7.7] w[+mC]=10xUAS(FRT.stop) myr::smGdP-V5-THS 10x UAS (FRT.stop) myr::smGdP-FLAG} su(Hw)attP1 | Bloomington *Drosophila* Stock Center | RRID:BDSC64085 | Referred to as "MCFO1" |
| Genetic reagent (*D. melanogaster*) | w[1118] P{y[+t7.7] w[+mC]=hsFLP G5.PEST}attP3; ifc^JS3/CyOTb; PBac{y[+mDint2] w[+mC]=10xUAS (FRT.stop)myr::smGd-PHA}VK0000 5 P{y[+t7.7] w[+mC]=10xUAS(FRT.stop) myr::smGdP-V5-THS 10x UAS(FRT.stop) myr::smGdP FLAG} su(Hw)attP1 | This study | Freely available from authors | MCFO1 line with ifc^JS3/ CyO Tb allele |
| Genetic reagent (*D. melanogaster*) | w[*]; P{y[+t7.7] w[+mC]=10XUAS-IVS-myr::GFP}attP2 | Bloomington *Drosophila* Stock Center | RRID:BDSC32197 | |
| Genetic reagent (*D. melanogaster*) | w[1118]; P{y[+t7.7] w[+mC]=GMR54H02-GAL4}attP2 | Bloomington *Drosophila* Stock Center | RRID:BDSC45784 | |
| Genetic reagent (*D. melanogaster*) | ifc-KO/CyO, P{2xTb[1]-RFP}; P{y[+t7.7] w[+mC]=GMR54H02-GAL4}attP2 | This study | Freely available from authors | Cortex glia GMR GAL4 line with ifcK0 allele |
| Genetic reagent (*D. melanogaster*) | w[1118]; P{y[+t7.7] w[+mC]=GMR86E01-GAL4}attP2 | Bloomington *Drosophila* Stock Center | RRID:BDSC45914 | Astrocyte-like glia GMR-GAL4 |
| Genetic reagent (*D. melanogaster*) | ifc-KO/CyO, P{2xTb[1]-RFP}; P{y[+t7.7] w[+mC] =GMR86E01-GAL4}attP2 | This study | Freely available from authors | Astrocyte-like glia GMR-GAL4 with copy of ifc-KO |

*Appendix 1 Continued on next page*

*Appendix 1 Continued*

| Reagent type (species) or resource | Designation | Source or reference | Identifiers | Additional information |
|---|---|---|---|---|
| Genetic reagent (*D. melanogaster*) | w[1118]; P{y[+t7.7] w[+mC]=GMR56F03-GAL4}attP2 | Bloomington *Drosophila* Stock Center | RRID:BDSC39157 | Ensheathing glia GMR-GAL4 |
| Genetic reagent (*D. melanogaster*) | ifc-KO/CyO, P{2xTb[1]-RFP}; P{y[+t7.7] w[+mC] =GMR56F03-GAL4}attP2 | This study | Freely available from authors | Ensheathing glia GMR-GAL4 with one copy of ifc-KO |
| Genetic reagent (*D. melanogaster*) | w[1118]; P{y[+t7.7] w[+mC]=GMR54C07-GAL4}attP2 | Bloomington *Drosophila* Stock Center | RRID:BDSC50472 | Subperineurial glia GMR-GAL4 |
| Genetic reagent (*D. melanogaster*) | ifc-KO/CyO, P{2xTb[1]-RFP}; P{y[+t7.7] w[+mC]=GMR54C07-GAL4}attP2 | This study | Freely available from authors | Subperineurial glia GMR-GAL4 with one copy of ifc-KO |
| Genetic reagent (*D. melanogaster*) | w[1118]; P{y[+t7.7] w[+mC]= GMR85G01-GAL4}attP2 | Bloomington *Drosophila* Stock Center | RRID:BDSC40436 | Perineurial glia GMR-GAL4 |
| Genetic reagent (*D. melanogaster*) | ifc-KO/CyO, P{2xTb[1]-RFP}; P{y[+t7.7] w[+mC]= GMR85G01-GAL4}attP2 | This study | Freely available from authors | Perineurial glia GMR-GAL4 with one copy of ifc-KO |
| Genetic reagent (*D. melanogaster*) | P{w[+mW.hs]=GawB}elav[C155] | Bloomington *Drosophila* Stock Center | RRID:BDSC458 | |
| Genetic reagent (*D. melanogaster*) | FlyFos019206(pRedFlp-Hgr) (ifc[27951]::2XTY1-SGFP-V5 -preTEV-BLRP-3XFLAG)dFRT | Vienna Dros. Research Center | RRID:VDRC318826 | |
| Genetic reagent (*D. melanogaster*) | P{VSH330794} | Vienna Dros. Research Center | RRID:VDRC330794 | RNAi line for ifc |
| Genetic reagent (*D. melanogaster*) | M{UAS-ifc-ORF-3xHA.attP}86Fb | FlyORF | FlyORF: F003887 (reference 72) | |
| Genetic reagent (*D. melanogaster*) | ifc^JS1 M{3xP3-RFP.attP}ZH-51D/CyO Tb | This study | Freely available from authors | V276D loss-of-function allele |
| Genetic reagent (*D. melanogaster*) | ifc^JS2 M{3xP3-RFP.attP}ZH-51D/CyO Tb | This study | Freely available from authors | G257S loss-of-function allele |
| Genetic reagent (*D. melanogaster*) | ifc^JS3 M{3xP3-RFP.attP}ZH-51D/CyO Tb | This study | Freely available from authors | W162* loss-of-function allele |
| Genetic reagent (*D. melanogaster*) | elav-GAL4[C155]; Repo-GAL80/CyO Tb | This study | Freely available from authors | |
| Genetic reagent (*D. melanogaster*) | ifc-KO/CyO Tb | Gift from Dr. Chih-Chiang Chan | *Jung et al., 2017* (Ref: 17) | |
| Antibody | Rabbit anti-FABP polyclonal | This study | Freely available from authors | 1:500 |
| Antibody | Rabbit anti-EBONY polyclonal | Gift from Dr. Haluk Lacin | RRID:AB_2314354 | 1:500 |
| Antibody | Mouse anti-REPO monoclonal | DSHB | RRID:AB_528448 | 1:100 |
| Antibody | Rat anti-ELAV monoclonal | DSHB | RRID:AB_528218 | 1:100 |
| Antibody | Mouse anti-CNX 99A monoclonal | DSHB | RRID:AB_2722011 | 1:20 |
| Antibody | Mouse anti-GOLGIN84 monoclonal | DSHB | RRID:AB_2722113 | 1:20 |
| Antibody | Goat anti-GOLGIN-245 polyclonal | DSHB | RRID:AB_2618260 | 1:500 |
| Antibody | Rabbit anti-ESYT polyclonal | Gift from Dr. Dion Dickman | | 1:300 |
| Antibody | GFP Antibody Dylight 488 Goat Polyclonal | Rockland (600-141-215) | RRID:AB_1961516 | 1:1000 |
| Antibody | Anti-LAMP1 antibody | Abcam (ab 30687) | RRID:AB_775973 | 1:500 (lyso-some marker) |

*Appendix 1 Continued on next page*

*Appendix 1 Continued*

| Reagent type (species) or resource | Designation | Source or reference | Identifiers | Additional information |
|---|---|---|---|---|
| Antibody | Cleaved Caspase-3 (Asp175) polyclonal | Cell Signaling Technology (#9661) | RRID:AB_2341188 | 1:400 |
| Antibody | Anti-HA antibody produced in rabbit | Millipore Sigma (H6908) | RRID:AB_260070 | 1:500 |
| Antibody | ANTI-FLAG antibody, Rat monoclonal | Millipore Sigma (SAB4200071) | RRID:AB_10603396 | 1:500 |
| Antibody | Chicken V5 Tag Polyclonal Antibody | Bethyl laboratories | RRID:AB_66741 | 1:500 |
| Antibody | Goat anti-Chicken IgY Alexa 488 | Invitrogen (A-11039) | RRID:AB_2534096 | 1:1000 |
| Antibody | Donkey anti-Rat Alexa 555 | Invitrogen (A48270) | RRID:AB_2896336 | 1:1000 |
| Antibody | Donkey Anti-Rat Cy5 | Jackson Immuno-Research | RRID:AB_2340672 | 1:1000 |
| Antibody | Donkey Anti-Goat IgG Cy5 | Jackson Immuno-Research | RRID:AB_2340415 | 1:1000 |
| Antibody | Donkey Anti-Mouse IgG Cy5 | Jackson Immuno-Research | RRID:AB_2340820 | 1:1000 |
| Antibody | Donkey Anti-Rabbit IgG Cy5 | Jackson Immuno-Research | RRID:AB_2340607 | 1:1000 |
| Chemical compound | BODIPY 493/503 | Invitrogen | D3922 | 1:200 |
| Software | ImageJ2 2.3.0/1.53q | NIH (https://imagej.net/) | RRID:SCR_003070 | |
| Software | ZEN Microscopy Software | Carl Zeiss AG; Jena, DEU | RRID:SCR_013672 | |
| Software | Photoshop 23.5.5 | Adobe; San Jose, CA | RRID:SCR_014199 | |
| Software | GraphPad Prism 10.2.2 | GraphPad; Boston, MA | RRID:SCR_002798 | |

