## [Editor Report · eLife Assessment]

This study on the loss of DEGS1 in the developing larval brain **convincingly** shows the accumulation of dihydroceramide in the CNS which induces severe alterations in the morphology of glial subtypes as well as a reduction in glial number. The localization of DEGS1/ifc primarily to the ER is also **compelling** and interesting, and the loss of DEGS1/ifc clearly drives ER expansion and reduces the levels of TGs. This is an **important** contribution to the role of lipid metabolism in neural development and disease.

---

## [Referee Report · Reviewer #1 (Public review)]

Summary:

Zhu et al., investigate the cellular defects in glia as a result of loss in DEGS1/ifc encoding the dihydroceramide desaturase. Using the strength of Drosophila and its vast genetic toolkit, they find that DEGS1/ifc is mainly expressed in glia and it's loss leads to profound neurodegeneration. This supports a role for DEGS1 in the developing larval brain as it safeguards proper CNS development. Loss of DEGS1/ifc leads to dihydroceramide accumulation in the CNS and induces alteration in the morphology of glial subtypes and a reduction in glial number. Cortex and ensheathing glia appeared swollen and accumulated internal membranes. Astrocyte-glia on the other hand displayed small cell bodies, reduced membrane extension and disrupted organization in the dorsal ventral nerve cord. They also found that DEGS1/ifc localizes primarily to the ER. Interestingly, the authors observed that loss of DEGS1/ifc drives ER expansion and reduced TGs and lipid droplet numbers. No effect on PC and PE and a slight increase in PS.

The conclusions of this paper are well supported by the data.

Strengths:

This is an interesting study that provides new insight into the role of ceramide metabolism in neurodegeneration.

The strength of the paper is the generation of LOF lines, the insertion of transgenes and the use of the UAS-GAL4/GAL80 system to assess the cell-autonomous effect of DEGS1/ifc loss in neurons and different glial subtypes during CNS development.

The imaging, immunofluorescence staining and EM of the larval brain and the use of the optical lobe and the nerve cord as a readout are very robust and nicely done.

Drosophila is a difficult model to perform core biochemistry and lipidomics, but the authors used the whole larvae and CNS to uncover global changes in mRNA levels related to lipogenesis and the unfolded protein responses, as well as specific lipid alterations upon DEGS1/ifc loss.

Weaknesses:

No major weaknesses identified.

Minor point: The authors performed lipidomics and RTqPCR on whole larvae and larval CNS which does not inform of any cell type-specific effects. Performing single-cell RNAseq on larval brains to tease apart the cell-type specific effect of DEGS1/ifc loss would be interesting to explore the future, but beyond the scope of the current study.

---

## [Referee Report · Reviewer #2 (Public review)]

Summary:

The manuscript by Zhu et al. describes phenotypes associated with the loss of the gene ifc using a Drosophila model. The authors suggest their findings are relevant to understanding the molecular underpinnings of a neurodegenerative disorder, HLD-18, which is caused by mutations in the human ortholog of ifc, DEGS1.

The work begins with the authors describing the role for ifc during fly larval brain development, demonstrating its function in regulating developmental timing, brain size, and ventral nerve cord elongation. Further mechanistic examination revealed that loss of ifc leads to depleted cellular ceramide levels as well as dihydroceramide accumulation, eventually causing defects in ER morphology and function. Importantly, the authors showed that ifc is predominantly expressed in glia and is critical for maintaining appropriate glial cell numbers and morphology. Many of the key phenotypes caused by the loss of fly ifc can be rescued by overexpression of human DEGS1 in glia, demonstrating the conserved nature of these proteins as well as the pathways they regulate. Interestingly, the authors discovered that the loss of lipid droplet formation in ifc mutant larvae within the cortex glia, presumably driving the deficits in glial wrapping around axons and subsequent neurodegeneration, potentially shedding light on mechanisms of HLD-18 and related disorders.

Strengths:

Overall, the manuscript is thorough in its analysis of ifc function and mechanism. The data images are high quality, the experiments are well controlled, and the writing is clear. There are, however, some concerns that need to be addressed prior to publication.

Weaknesses:

The authors adequately addressed the previously indicated weaknesses, and no new weaknesses have been identified.

---

## [Referee Report · Reviewer #3 (Public review)]

Summary:

In this manuscript, the authors report three novel ifc alleles: ifc[js1], ifc[js2], and ifc[js3]. ifc[js1] and ifc[js2] encode missense mutations, V276D and G257S, respectively. ifc[js3] encodes a nonsense mutation, W162*. These alleles exhibit multiple phenotypes, including delayed progression to the late-third larval instar stage, reduced brain size, elongation of the ventral nerve cord, axonal swelling, and lethality during late larval or early pupal stages.

Further characterization of these alleles the authors reveals that ifc is predominantly expressed in glia and localizes to the endoplasmic reticulum (ER). The expression of ifc gene governs glial morphology and survival. Expression of fly ifc cDNA or human DEGS1 cDNA specifically in glia, but not neurons, rescues the CNS phenotypes of ifc mutants, indicating a crucial role for ifc in glial cells and its evolutionary conservation. Loss of ifc results in ER expansion and loss of lipid droplets in cortex glia. Additionally, loss of ifc leads to ceramide depletion and accumulation of dihydroceramide. Moreover, it increases the saturation levels of triacylglycerols and membrane phospholipids. Finally, the reduction of dihydroceramide synthesis suppresses the CNS phenotypes associated with ifc mutations, indicating the key role of dihydroceramide in causing ifc LOF defects.

Strengths:

This manuscript unveils several intriguing and novel phenotypes of ifc loss-of-function in glia. The experiments are meticulously planned and executed, with the data strongly supporting their conclusions.

---

## [Author Response]

The following is the authors’ response to the original reviews.

**Reviewer #1 (Public Review):**
**Summary**: Zhu et al., investigate the cellular defects in glia as a result of loss in DEGS1/ifc encoding the dihydroceramide desaturase. Using the strength of Drosophila and its vast genetic toolkit, they find that DEGS1/ifc is mainly expressed in glia and its loss leads to profound neurodegeneration. This supports a role for DEGS1 in the developing larval brain as it safeguards proper CNS development. Loss of DEGS1/ifc leads to dihydroceramide accumulation in the CNS and induces alteration in the morphology of glial subtypes and a reduction in glial number. Cortex and ensheathing glia appeared swollen and accumulated internal membranes. Astrocyte-glia on the other hand displayed small cell bodies, reduced membrane extension and disrupted organization in the dorsal ventral nerve cord. They also found that DEGS1/ifc localizes primarily to the ER. Interestingly, the authors observed that loss of DEGS1/ifc drives ER expansion and reduced TGs and lipid droplet numbers. No effect on PC and PE and a slight increase in PS.The conclusions of this paper are well supported by the data. The study could be further strengthened by a few additional controls and/or analyses.Strengths:This is an interesting study that provides new insight into the role of ceramide metabolism in neurodegeneration.The strength of the paper is the generation of LOF lines, the insertion of transgenes and the use of the UAS-GAL4/GAL80 system to assess the cell-autonomous effect of DEGS1/ifc loss in neurons and different glial subtypes during CNS development.The imaging, immunofluorescence staining and EM of the larval brain and the use of the optical lobe and the nerve cord as a readout are very robust and nicely done.Drosophila is a difficult model to perform core biochemistry and lipidomics but the authors used the whole larvae and CNS to uncover global changes in mRNA levels related to lipogenesis and the unfolded protein responses as well as specific lipid alterations upon DEGS1/ifc loss.Weaknesses:(1) The authors performed lipidomics and RTqPCR on whole larvae and larval CNS from which it is impossible to define the cell type-specific effects. Ideally, this could be further supported by performing single cell RNAseq on larval brains to tease apart the cell-type specific effect of DEGS1/ifc loss.

We agree that using scRNAseq or pairing FACS-sorting of individual glial subtypes with bulk RNAseq would help tease apart the cell-type specific effects of DEGS1/ifc loss on glial cells. At this time, however, this approach extends beyond the scope of the current paper and means of the lab.

(2) It's clear from the data that the accumulation of dihydroceramide in the ER triggers ER expansion but it remains unclear how or why this happens. Additionally, the authors assume that, because of the reduction in LD numbers, that the source of fatty acids comes from the LDs. But there is no data testing this directly.

As CERT, the protein that transports ceramide from the ER to the Golgi, is far more efficient at transporting ceramide than dihydroceramide, we speculate that dihydroceramide accumulates in the ER due to inefficient transport from the ER to the Golgi by CERT. We state this model more explicitly in the results under the subheading “Reduction of dihydroceramide synthesis suppresses the ifc CNS phenotype”.

We agree with the point on lipid droplet. We observe a correlation, not a causation, between reduction of lipid droplets and a large expansion of ER membrane. We have tried to clarify the text in the last paragraph of the discussion to make this point more clearly. See also response to reviewer 2 point 3.

(3) The authors performed a beautiful EMS screen identifying several LOF alleles in ifc. However, the authors decided to only use KO/ifcJS3. The paper could be strengthened if the authors could replicate some of the key findings in additional fly lines.

We agree. We replicated the observed cortex glia swelling, ER expansion in cortex glia, and observed increase in neuronal cell death markers in late-third instar larvae mutant for either the ifcjs1 or ifcjs2 allele. These data are now provided as Supplementary Figure 7.

(4) The authors use M{3xP3-RFP.attP}ZH-51D transgene as a general glial marker. However, it would be advised to show the % overlap between the glial marker and the RFP since a lot of cells are green positive but not per se RFP positive and vice versa.

We visually reexamined the expression of the 3xP3 RFP transgene relative to FABP labeling for cortex glia, Ebony for astrocyte-like glia, and the Myr-GFP transgene driven by glial-subtype specific GAL4 driver lines for perineurial, subperineurial, and ensheathing glia. We note that RFP localizes to the nucleus cytoplasm while FABP and Ebony localize to the cytoplasm and Myr-GFP to the cell membrane. Thus, an observed lack of overlap of expression between RFP and the other markers can arise to differential localization of the two markers in the same cells see, for example, Fig. S2D where Myr-GFP expression in the nuclear envelope encircles that of RFP in the nucleus. Through visual inspection of five larval-brain complexes for each glial subtype marker, we found that essentially all cortex, SPG, and ensheathing glia expressed RFP. Similarly, nearly all astrocyte-like glia also expressed RFP, but they expressed RFP at significantly lower levels than that observed for cortex, SPG, or ensheathing glia. This analysis also confirmed that most perineurial glia do not express RFP. The 3xP3 M{3xP3-RFP.attP}ZH-51D transgene then labels most glia in the Drosophila CNS. We have added text to Supplementary Figure 2 noting the above observations as to which glial cells express RFP.

(5) The authors indicate that other 3xP3 RFP and GFP transgenes at other genomic locations also label most glia in the CNS. Do they have a preferential overlap with the different glial subtypes?

We assessed three different types of 3xP3 RFP and GFP transgenes: M{3xP3RFP.attp} transgenes (n=4), Mi{GFP[E.3xP3]=ET1} transgenes (n=3), and

Tl{GFP[3xP3.cLa]=CRIMIC.TG4} transgenes (n>6). All labeled cortex glia, but different lines exhibited differential labeling of astrocyte and ensheathing glia. These data are now included as Supplementary Figure 3.

**Reviewer #2 (Public Review):**
Summary:The manuscript by Zhu et al. describes phenotypes associated with the loss of the gene ifc using a Drosophila model. The authors suggest their findings are relevant to understanding the molecular underpinnings of a neurodegenerative disorder, HLD-18, which is caused by mutations in the human ortholog of ifc, DEGS1.The work begins with the authors describing the role for ifc during fly larval brain development, demonstrating its function in regulating developmental timing, brain size, and ventral nerve cord elongation. Further mechanistic examination revealed that loss of ifc leads to depleted cellular ceramide levels as well as dihydroceramide accumulation, eventually causing defects in ER morphology and function. Importantly, the authors showed that ifc is predominantly expressed in glia and is critical for maintaining appropriate glial cell numbers and morphology. Many of the key phenotypes caused by the loss of fly ifc can be rescued by overexpression of human DEGS1 in glia, demonstrating the conserved nature of these proteins as well as the pathways they regulate. Interestingly, the authors discovered that the loss of lipid droplet formation in ifc mutant larvae within the cortex glia, presumably driving the deficits in glial wrapping around axons and subsequent neurodegeneration, potentially shedding light on mechanisms of HLD-18 and related disorders.Strengths:Overall, the manuscript is thorough in its analysis of ifc function and mechanism. The data images are high quality, the experiments are well controlled, and the writing is clear.Weaknesses:(1) The authors clearly demonstrated a reduction in number of glia in the larval brains of ifc mutant flies. What remains unclear is whether ifc loss leads to glial apoptosis or a failure for glia to proliferate during development. The authors should distinguish between these two hypotheses using apoptotic markers and cell proliferation markers in glia.

To address this point, we used phospho-histone H3 to assess mitotic index in the thoracic CNS of wild-type versus ifc mutant late third instar larvae and found a mild, but significant reduction in mitotic index in ifc mutant relative to wild-type nerve cords. We also assessed the ability of glial-specific expression of the potent anti-apoptotic gene p35 to rescue the observed loss of cortex glia phenotype in the thoracic region of the CNS of otherwise ifc mutant larvae and observed a clear increase in cortex glia in the presence versus the absence of glial-specific p35 expression (p<3 x 10-4). These data are now provided as Supplementary Figure S8 in the paper and referred to on page 8.

(2) It is surprising that human DEGS1 expression in glia rescues the noted phenotypes despite the different preference for sphingoid backbone between flies and mammals. Though human DEGS1 rescued the glial phenotypes described, can animal lethality be rescued by glial expression of human DEGS1? Are there longer-term effects of loss of ifc that cannot be compensated by the overexpression of human DEGS1 in glia (age-dependent neurodegeneration, etc.)?

We note explicitly that while glial expression of human DEGS1 does provide rescuing activity, it only partially rescues the ifc mutant CNS phenotype in contrast to glial expression of Drosophila ifc, which fully rescues this phenotype. Thus, the relative activity of human DEGS1 is far below that of Drosophila ifc when assayed in flies. To quantify the functional difference between the two transgenes, we assessed the ability of glial expression of fly ifc or of human DEGS1 to rescue the lethality of otherwise ifc mutant larvae: Glial expression of ifc was sufficient to rescue the adult viability of 57.9% of ifc mutant flies based on expected Mendelian ratios (n=2452), whereas glial expression of DEGS1 was sufficient to rescue just 3.9% of ifc mutant flies (n=1303), uncovering a ~15-fold difference in the ability of the two transgenes to rescue the lethality of otherwise ifc mutant flies. In the absence of either transgene, no ifc mutant larvae reached adulthood (n=1030). These data are now provided in the text on page 9 of the revised manuscript.

(3) The mechanistic link between the loss of ifc and lipid droplet defects is missing. How do defects in ceramide metabolism alter triglyceride utilization and storage? While the author's argument that the loss of lipid droplets in larval glia will lead to defects in neuronal ensheathment, a discussion of how this is linked to ceramides needs to be added.

We have revised the text to address this point. We speculate that the apparent increased demand for membrane phospholipid synthesis may drive the depletion of lipid droplets, providing a link to ifc function and ceramides. Below we provide the rewritten last paragraph; the underlined section is the new text.

“The expansion of ER membranes coupled with loss of lipid droplets in ifc mutant larvae suggests that the apparent demand for increased membrane phospholipid synthesis may drive lipid droplet depletion, as lipid droplet catabolism can release free fatty acids to serve as substrates for lipid synthesis. At some point, the depletion of lipid droplets, and perhaps free fatty acids as well, would be expected to exhaust the ability of cortex glia to produce additional membrane phospholipids required for fully enwrapping neuronal cell bodies. Under wild-type conditions, many lipid droplets are present in cortex glia during the rapid phase of neurogenesis that occurs in larvae. During this phase, lipid droplets likely support the ability of cortex glia to generate large quantities of membrane lipids to drive membrane growth needed to ensheathe newly born neurons. Supporting this idea, lipid droplets disappear in the adult Drosophila CNS when neurogenesis is complete and cortex glia remodeling stops. We speculate that lipid droplet loss in ifc mutant larvae contributes to the inability of cortex glia to enwrap neuronal cell bodies. Prior work on lipid droplets in flies has focused on stress-induced lipid droplets generated in glia and their protective or deleterious roles in the nervous system. Work in mice and humans has found that more lipid droplets are often associated with the pathogenesis of neurodegenerative diseases, but our work correlates lipid droplet loss with CNS defects. In the future, it will be important to determine how lipid droplets impact nervous system development and disease.”

(4) On page 10, the authors use the words "strong" and "weak" to describe where ifc is expressed. Since the use of T2A-GAL4 alleles in examining gene expression is unable to delineate the amount of gene expression from a locus, the terms "broad" and "sparse" labeling (or similar terms) should be used instead.

The ifc T2A-GAL4 insert in the ifc locus reports on the transcription of the gene. We agree that GAL4 system will not reflect amount of gene expression differences when the expression levels are not dramatically different. However, when the expression levels differ dramatically, as in our case, GAL4 system can reflect this difference in the expression of a reporter gene. We reworded this section to suggest that ifc is transcribed at higher levels in glia as compared to neurons. We can’t use sparse or broad, as ifc is expressed in all, or at least in most, glia and neurons. The new text is as follows:” Using this approach, we observed strong nRFP expression in all glial cells (Figures 4D and S10A) and modest nRFP expression in all neurons (Figures 4E and S10B), suggesting ifc is transcribed at higher levels in glial cells than neurons in the larval CNS.”

**Reviewer #3 (Public Review):**
Summary:In this manuscript, the authors report three novel ifc alleles: ifc[js1], ifc[js2], and ifc[js3]. ifc[js1] and ifc[js2] encode missense mutations, V276D and G257S, respectively. ifc[js3] encodes a nonsense mutation, W162*. These alleles exhibit multiple phenotypes, including delayed progression to the late-third larval instar stage, reduced brain size, elongation of the ventral nerve cord, axonal swelling, and lethality during late larval or early pupal stages.Further characterization of these alleles the authors reveals that ifc is predominantly expressed in glia and localizes to the endoplasmic reticulum (ER). The expression of ifc gene governs glial morphology and survival. Expression of fly ifc cDNA or human DEGS1 cDNA specifically in glia, but not neurons, rescues the CNS phenotypes of ifc mutants, indicating a crucial role for ifc in glial cells and its evolutionary conservation. Loss of ifc results in ER expansion and loss of lipid droplets in cortex glia. Additionally, loss of ifc leads to ceramide depletion and accumulation of dihydroceramide. Moreover, it increases the saturation levels of triacylglycerols and membrane phospholipids. Finally, the reduction of dihydroceramide synthesis suppresses the CNS phenotypes associated with ifc mutations, indicating the key role of dihydroceramide in causing ifc LOF defects.Strengths:This manuscript unveils several intriguing and novel phenotypes of ifc loss-of-function in glia. The experiments are meticulously planned and executed, with the data strongly supporting their conclusions.Weaknesses:I didn't find any obvious weakness.
**Reviewer #1 (Recommendations For The Authors):**
Additional minor comments below:(1) The authors state that TGs are the building blocks of membrane phospholipids. This is not exactly true. The breakdown of TGs can result in free FAs which can be used for membrane phospholipid synthesis. Also, membrane phospholipids can also be generated from free FAs that were never in TGs.

To address this point, we have reworked a number of sentences in the text. On page 12 we reworded two small sections to the following:

“In the CNS, lipid droplets form primarily in cortex glia[29] and are thought to contribute to membrane lipid synthesis through their catabolism into free fatty acids versus acting as an energy source in the brain.[41] Consistent with the possibility that increased membrane lipid synthesis drives lipid droplet reduction, RNA-seq assays of dissected nerve cords revealed that loss of ifc drove transcriptional upregulation of genes that promote membrane lipid biogenesis”

As TG breakdown results in free fatty acids that can be used for membrane phospholipid synthesis, we asked if changes in TG levels and saturation were reflected in the levels or saturation of the membrane phospholipids phosphatidylcholine (PC), phosphatidylethanolamine (PE), and phosphatidylserine (PS).

(2) Figure 5J what does the dotted line indicate? Please specify in the figure legend or remove it.

We have added the following text in the figure legend: Dotted line indicates a log2 fold change of 0.5 in the treatment group compared to the control group.

(3) The text for your graphs is hard to read. Please make the font larger.

We have increased font size to enhance the readability of the figures.

(4) The authors mentioned that driving *ifc* expression in neurons rescues the phenotypes (ref 17). While the glial-specific role presented in this study is robust. I think some readers would appreciate some discussion of this study in light of the data presented here.

We have added the below text on page 10 to address this point.

“Results of our gene rescue experiments conflict with a prior study on ifc in which expression of ifc in neurons was found to rescue the ifc phenotype. In this context, we note that elav-GAL4 drives UASlinked transgene expression not just in neurons, but also in glia at appreciable levels, and thus needs to be paired with repo-GAL80 to restrict GAL4-mediated gene expression to neurons. Thus, “off-target” expression in glial cells may account for the discrepant results. It is, however, more difficult to reconcile how neuronal or glial expression of ifc would rescue the observed lethality of the ifc-KO chromosome given the presence additional lethal mutations in the 21E2 region of the second chromosome.”

(5) While the analysis of fatty acid saturation is experimentally well done. I'm not really sure what the significance of this data is.

We included this information as a reference for future analysis of additional genes in the ceramide biogenesis pathway, as we expect that alteration of the levels and saturation levels of PE, PC, and PS in cell membranes may underlie key changes in the biophysical properties of glial cell membranes and their ability to enwrap or infiltrate their targets. Thus, we expect the significance of these data to grow as more work is done on additional members of the ceramide pathway in the nervous system in flies and other systems.

**Reviewer #2 (Recommendations For The Authors):**
(1) There is a typo at the top of page 11: "internal membranes and fail enwrap neurons" is missing the word "to" before "enwrap"

The typo was fixed.

(2) PMID: 36718090 should be included in the discussion of SPT and ORMDL complex in human disease.

The reference was added.

**Reviewer #3 (Recommendations For The Authors):**
In this manuscript, the authors report three novel ifc alleles: ifc[js1], ifc[js2], and ifc[js3]. ifc[js1] and ifc[js2] encode missense mutations, V276D and G257S, respectively. ifc[js3] encodes a nonsense mutation, W162*. These alleles exhibit multiple phenotypes, including delayed progression to the late-third larval instar stage, reduced brain size, elongation of the ventral nerve cord, axonal swelling, and lethality during late larval or early pupal stages.Further characterization of these alleles the authors reveals that ifc is predominantly expressed in glia and localizes to the endoplasmic reticulum (ER). The expression of ifc gene governs glial morphology and survival. Expression of fly ifc cDNA or human DEGS1 cDNA specifically in glia, but not neurons, rescues the CNS phenotypes of ifc mutants, indicating a crucial role for ifc in glial cells and its evolutionary conservation. Loss of ifc results in ER expansion and loss of lipid droplets in cortex glia. Additionally, loss of ifc leads to ceramide depletion and accumulation of dihydroceramide. Moreover, it increases the saturation levels of triacylglycerols and membrane phospholipids. Finally, the reduction of dihydroceramide synthesis suppresses the CNS phenotypes associated with ifc mutations, indicating the key role of dihydroceramide in causing ifc LOF defects.In summary, this manuscript unveils several intriguing and novel phenotypes of ifc loss-of-function in glia. The experiments are meticulously planned and executed, with the data strongly supporting their conclusions. I have no additional comments and fully support the publication of this manuscript in eLife.

The authors also note that they added one paragraph to the discussion that addresses the possibility that the increased detection of cell death markers could arise due to the inability of glial cells to remove cellular debris. The text of this paragraph is provided below:

We note that cortex glia are the major phagocytic cell of the CNS and phagocytose neurons targeted for apoptosis as part of the normal developmental process.23-26 Thus, while we favor the model that ifc triggers neuronal cell death due to glial dysfunction, it is also possible that increased detection of dying neurons arises due at least in part to a decreased ability of cortex glia to clear dying neurons from the CNS. At present, the large number of neurons that undergo developmentally programmed cell death combined with the significant disruption to brain and ventral nerve cord morphology caused by loss of ifc function render this question difficult to address.Additional evidence does, however, support the idea that loss of ifc function drives excess neuronal cell death: Clonal analysis in the fly eye reveals that loss of ifc drives photoreceptor neuron degeneration17, indicating that loss of ifc function drives neuronal cell death; cortex-glia specific depletion of CPES, which acts downstream of ifc, disrupts neuronal function and induces photosensitive epilepsy in flies59, indicating that genes in the ceramide pathway can act nonautonomously in glia to regulate neuronal function; recent genetic studies reveal that other glial cells can compensate for impaired cortex glial cell function by phagocytosing dying neurons62, and we observe that the cell membranes of subperineurial glia enwrap dying neurons in ifc mutant larvae (Fig. S14), consistent with similar compensation occurring in this background, and in humans, loss of function mutations in DEGS1 cause neurodegeneration.7-9 Clearly, future work is required to address this question for ifc/DEGS1 and perhaps other members of the ceramide biogenesis pathway.